# Controllable Data Generation with Hierarchical Neural Representations

**Sheyang Tang** [1]  **Xiaoyu Xu** ✉ [1]  **Jiayan Qiu** [2]  **Zhou Wang** [1]

## Abstract

Implicit Neural Representations (INRs) represent data as continuous functions using the parameters of a neural network, where data information is encoded in the parameter space. Therefore, modeling the distribution of such parameters is crucial for building generalizable INRs. Existing approaches learn a joint distribution of these parameters via a latent vector to generate new data, but such a flat latent often fails to capture the inherent hierarchical structure of the parameter space, leading to entangled data semantics and limited control over the generation process. Here, we propose a **C**ontrollable **H**ierarchical **I**mplicit **N**eural **R**epresentation (**CHINR**) framework, which explicitly models conditional dependencies across layers in the parameter space. Our method consists of two stages: In Stage-1, we construct a Layers-of-Experts (LoE) network, where each layer modulates distinct semantics through a unique latent vector, enabling disentangled and expressive representations. In Stage-2, we introduce a Hierarchical Conditional Diffusion Model (HCDM) to capture conditional dependencies across layers, allowing for controllable and hierarchical data generation at various semantic granularities. Extensive experiments across different modalities demonstrate that CHINR improves generalizability and offers flexible hierarchical control over the generated content.

## 1. Introduction

Implicit Neural Representations (INRs) are powerful tools to represent complex data as continuous functions with neural networks (Tancik et al., 2020; Mildenhall et al., 2021;

[1]Department of Electrical and Computer Engineering, University of Waterloo, Waterloo, Canada [2]School of Computing and Mathematical Sciences, University of Leicester, Leicester, United Kindom. Correspondence to: Xiaoyu Xu <x423xu@uwaterloo.ca>.

*Proceedings of the 42nd International Conference on Machine Learning*, Vancouver, Canada. PMLR 267, 2025. Copyright 2025 by the author(s).

You et al., 2024), offering compact and universal representations across diverse data modalities such as audio (Su et al., 2022), images (Sitzmann et al., 2020; Dupont et al., 2021), videos (Chen et al., 2021a; 2023), and 3D volumes (Mildenhall et al., 2021; Zhao et al., 2022; Michalkiewicz et al., 2019). By modeling data as functions $f_{\boldsymbol{\theta}} : \mathcal{X} \to \mathcal{F}$, with $\mathcal{X}$ and $\mathcal{F}$ being the input (e.g., pixel coordinates) and output (e.g., RGB values), INRs implicitly encode data as a hidden manifold within the parameter space of $\boldsymbol{\theta}$, capturing the underlying structure of the data. By modeling the distribution of parameters $p(\boldsymbol{\theta})$, generative INRs present promising potentials for universal data generation (Dupont et al., 2022c;a; Bauer et al., 2023; You et al., 2024). Nevertheless, two fundamental questions have long been overlooked:

*How are these parameters related to data semantics, and hence how to control the parameters to generate expected semantics?*

Addressing this gap is critical for advancing the controllability of INR-based generative frameworks. Notably, we observe that INRs (e.g., SIREN (Sitzmann et al., 2020)) naturally exhibit a hierarchical structure, where each layer progressively expands the representational capacity of the model (Yüce et al., 2022). This expansion is intrinsically linked to the frequency basis of INRs: earlier layers capture coarse-grained features, while later layers progressively refine fine-grained details (Section 2.2 provides detailed analysis). As shown in Fig. 1, this progression aligns with the semantic abstraction hierarchy in data. For example, in facial images, hierarchical semantics are reflected in progressively detailed facial attributes, such as overall facial shape, expression, and shape of eyes. This connection between INR's hierarchical structure and data semantic abstraction offers a natural pathway to achieve hierarchical control.

However, existing generative INR approaches (Dupont et al., 2022a; Bauer et al., 2023; You et al., 2024) overlook this hierarchy, instead modeling the joint distribution of flattened INR parameters, either directly as $p(\boldsymbol{\theta})$ or through the distribution of condensed latents. It ignores the correspondence between the INRs' expanded representation capacity and the hierarchical semantics in the data. This misalignment brings two challenges: (1) Control over the generated content is limited. While generative models, such as latent diffusion, are employed with INRs to generate new

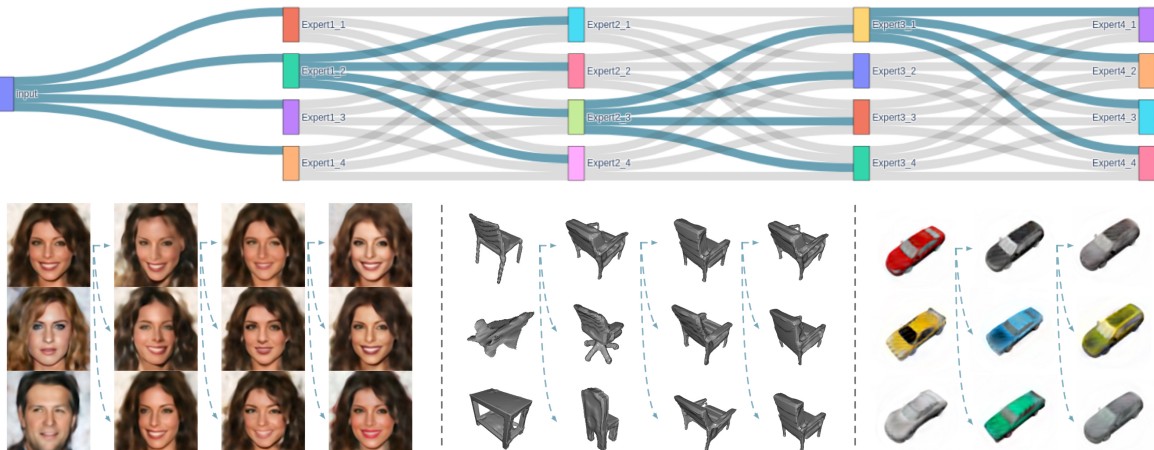

*Figure 1.* Hierarchical generation of universal data modality. *Top*: expanded representational capabilities of a Layers-of-Experts INR model. *Bottom*: aligning these capabilities with hierarchical data patterns enables precise control over the generation process. Each column presents generated samples resulting from a divergence in routing at a specific layer, with arrows indicating the shared routing in the preceding layers.

data, they cannot link the sampled noise to the expected semantics in the output. (2) The generalizability of INRs to unseen data is impaired. The joint distribution learning encodes the entangled semantics together into one flat latent, where the co-occurrence of certain semantics is inevitable. Therefore, the diversity of the generated data is limited.

To address this gap, we propose a **C**ontrollable **H**ierarchical **I**mplicit **N**eural **R**epresentation (**CHINR**) framework that investigates the hierarchical structure of INRs, promoting layer-wise control in the generation process. Our method starts by training a collection of INRs on a dataset. Each layer of an INR is parameterized as a Mixture-of-Experts (MoE) layer to increase expressivity, where a set of expert weights and latents are learned. The experts at each layer are shared across the dataset, while the latents are data-specific, routing the data flow and modulating the contribution of experts. Layers of MoE are cascaded to form an INR, which we call a Layers-of-Experts (LoE) network. Consequently, a LoE with $L$ layers will have $L$ latents adapted to the fitted data, effectively capturing and relating its complex patterns with layers of latents. By modeling the conditional dependencies of these layer-wise latents with a hierarchical conditional diffusion model (HCDM), we maintain the hierarchical structure of INRs. This unlocks a controllable generation process, aligning the layers in INRs with the hierarchies of data semantics for the first time. As illustrated in Fig. 1, the data flow for a generation process resembles a tree-like structure: the routing in the next layer is constrained by the paths in previous layers, allowing full control of where a different routing strategy should be explored. Early deviations in routing lead to significant semantic differences in the generated content, while a later deviation

results in minor differences in details.

The contributions of our paper are summarized as follows:

- We are the first to achieve hierarchical control in generative INRs by modeling hierarchical and conditional dependencies in INR parameters. This enables layer-wise, precise control over data semantics during generation.

- We model the INR as a LoE framework to enhance expressivity. This design aligns the inherent hierarchy of data semantics with INR's expanded representation capacity, enabling the generation of diverse data.

- The proposed CHINR shows broad versatility across modalities, highlighting the broad applicability of hierarchical control and conditional dependency modeling to data semantics with inherent hierarchies.

- We conduct extensive experiments across various modalities, demonstrating CHINR's superior performance in reconstruction and generation metrics.

## 2. Background

In this section, we introduce INR and generative INRs, while highlighting their connections to our proposed approach. We also analyze the inherent hierarchy in INR's parameters.

### 2.1. Implicit Neural Representation & Generative INRs

Implicit Neural Representations (INRs) parameterize data such as audio, images, video, and 3D voxels as mappings from coordinates to signals, enabling a unified framework for various data modalities (Genova et al., 2019a;b; Xie

et al., 2022). Remarkable progress has been made to enhance the representation quality, efficiency and compactness for audio (Zuiderveld et al., 2021; Luo et al., 2022; Su et al., 2022; Lanzendörfer & Wattenhofer), images (Sitzmann et al., 2020; Fathony et al., 2020; Chen et al., 2021b; Xu et al., 2022a; Saragadam et al., 2023; Yu et al., 2024; Xu et al., 2024b; 2022b; Qiu et al., 2021), 3D contents (Mildenhall et al., 2021; Barron et al., 2021; Tiwari et al., 2022; Ortiz et al., 2022; Zhao et al., 2022; Ruan et al., 2024; Qiu et al., 2020; 2019; Yang et al., 2020), and videos (Chen et al., 2021a; Li et al., 2022; Yan et al., 2024). Despite performing well on different modalities, INRs struggle to generalize to multiple and unseen data, as each instance is typically overfitted with a separate MLP. To address this, two key strategies have emerged: (1) learning content-specific input features (Yu et al., 2021; Hu et al., 2023; Lazova et al., 2023) and (2) modulating or customizing network parameters with latents or hypernetworks (Mehta et al., 2021; Wang et al., 2022; Dupont et al., 2022b; Kim et al., 2023; Xu et al., 2024a). Generative models (Goodfellow et al., 2014; Ho et al., 2020) further extend INR's capability to generate new data. GRAF (Schwarz et al., 2020) and GIRAFFE (Niemeyer & Geiger, 2021) generate shape and appearance codes from noise, which are combined with coordinates to construct scenes. Erkoç et al. (2023) use a diffusion model to generate INR weights. Dupont et al. (2022c); Du et al. (2021); Koyuncu et al. (2023) train hypernetworks to generate INR parameters. Dupont et al. (2022a); Bauer et al. (2023); Park et al. (2024) employ a two-stage framework to learn the distribution of latents that map to or modulate INRs, and generate new content by sampling in the latent space. mNIF (You et al., 2024) further enhances the expressivity of INR via model averaging. These methods essentially model the distribution of INR parameters $p(\boldsymbol{\theta})$ by learning the latent distributions $p(\boldsymbol{h})$, but fail to capture the layer-wise hierarchical structure of INR parameters (Section 2.2), limiting their ability to accurately model distributions and control generation. Building on the latent modulation approach, we introduce a hierarchical conditional diffusion model, capturing dependencies between layer-wise latents for improved generalization and control.

### 2.2. Hierarchy Analysis of INR

In this section, we review the INR architecture and analyze its inherent hierarchical representation ability. Using SIREN (Sitzmann et al., 2020) as an example, a two-layer SIREN is generally formulated as:

$$f_{\boldsymbol{\theta}}(\mathbf{x}) = \mathbf{W}_2 \sin(\mathbf{W}_1 \cdot \gamma(\mathbf{x})), \boldsymbol{\theta} = [\mathbf{W}_1, \mathbf{W}_2], \quad (1)$$

where $\gamma(\mathbf{x}) = \sin(\Omega \cdot \mathbf{x}), \Omega \in \mathbb{R}^{c_1 \times c_{in}}$ denotes positional embedding of coordinates $\mathbf{x}$, $\mathbf{W}_2 \in \mathbb{R}^{c_{out} \times c_2}$, $\mathbf{W}_1 \in \mathbb{R}^{c_2 \times c_1}$ denote the parameters of each layer. The bias is omitted for simplification. From the perspective of

Fourier Transform, the input frequency domain $\Omega$ is composed of $c_1$ frequency basis, $\Omega = [\Omega_0, \Omega_2, \cdots, \Omega_{c_1-1}]$. According to the Tancik et al. (2020) and Yüce et al. (2022), an MLP layer with periodic activation $\sin(\cdot)$ only expands the input frequency basis in a sparse and limited bandwidth. The equation 1 can be reformulated as:

$$f_{\boldsymbol{\theta}}(\mathbf{x}) = \sum_{w' \in \mathcal{H}(\Omega)} \alpha_{w'} \sin(w' \cdot \mathbf{x}),$$

$$\alpha_{w'} \propto \mathbf{W}_2 \cdot \prod_{c=0}^{c_1-1} \mathcal{J}_{s_c}(\mathbf{W}_{1[\cdot,c]}) \quad (2)$$

$$\mathcal{H}(\Omega) \subseteq \{\sum_{c=0}^{c_1-1} \beta_c \Omega_c | \beta_c \in \mathbb{Z} \ \& \ \sum_{c=0}^{c_1-1} \beta_c \leq K\},$$

where $\mathcal{J}_{s_c}$ denotes a Bessel function, $\mathbf{W}_{1[\cdot,c]}$ denotes the column $c$ of $\mathbf{W}_1$. The Eq. (2) reveals the properties of each $\sin(\cdot)$ activated INR layer in two aspects. First, the output spectrum of layer 2, i.e. $\alpha_{w'}$, is dependent on the spectrum of layer 1, determined by $\mathbf{W}_1$; Second, the output frequency domain $\mathcal{H}(\Omega)$ is sparse since $\beta_c$ is an integer, so $\mathcal{H}(\Omega)$ only covers sparse frequency space spanned by the basis $\{\beta_c \Omega_c\}$. These suggest that INR layers' spectrum and frequency basis inherently exhibit a sparse and hierarchical structure, encoded by $\boldsymbol{\theta}$, which extends to their representation ability. Latent-modulation approaches like mNIF (You et al., 2024), which model the parameter distribution $p(\boldsymbol{\theta})$ with the surrogate task of modeling the latent distribution $p(\boldsymbol{h})$, overlook the hierarchy in $\boldsymbol{h}$ transferred from $\boldsymbol{\theta}$. Ignoring this hierarchy leads to reduced expressivity and generalizability, and limited control over the generation process.

## 3. Proposed Method

Our method uses a two-stage framework to align the hierarchy of data semantic and INR's expanded representation ability, as shown in Fig. 2. In Stage-1, we train individual INRs to fit a target dataset. In Stage-2, we use generative models to learn weight distributions to generate new data.

Directly modeling INR's weight distribution brings three challenges: (1) independently trained INRs make it hard to extract shared information for distribution learning, (2) the high dimensionality of raw weights makes distribution modeling highly challenging, and (3) it ignores the hierarchical structure of INRs. To address them, we configure INR as a Layers-of-Experts (LoE) network, where each layer contains a set of shared expert weights and an instance-specific latent. As shown in Fig. 2 *left*, the inference process builds each INR layer by layer, combining experts with corresponding latents. This structure captures shared information through experts, simplifies distribution learning by focusing on latents, and explicitly models conditional dependencies within the hierarchical structure of INRs. In the following

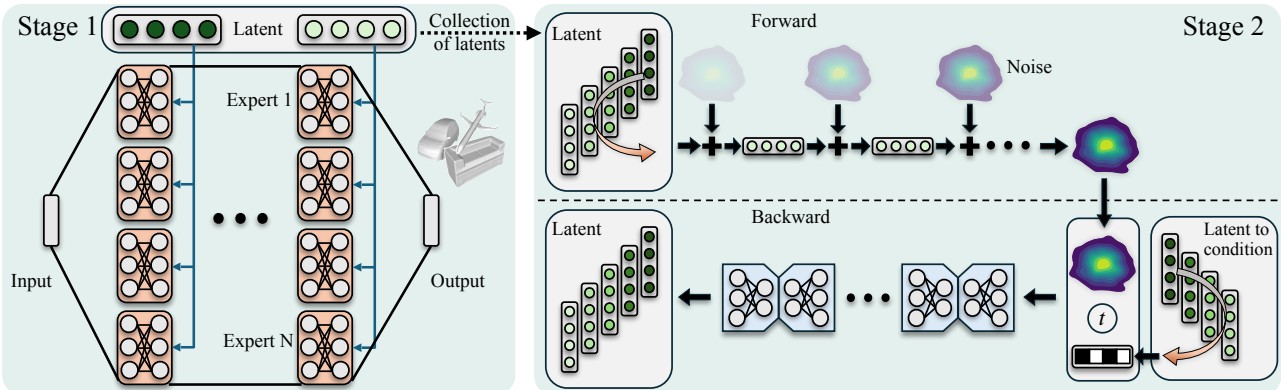

*Figure 2.* CHINR consists of two stages. In Stage-1, a Layer-of-Experts (LoE) model represents data with instance-specific latents and shared experts. The latent at each layer (shaded differently) modulates the mixture of experts at that layer. Stage-2 introduces a Hierarchical Conditional Diffusion Model (HCDM) to learn layer-wise conditional distributions of latents. At inference, we sample latents according to the conditional chain to achieve hierarchical control.

sections, we first define the LoE structure and learning task, followed by detailed explanations of the two stages.

### 3.1. Problem Statement

Suppose an INR $f_{\boldsymbol{\theta}}$ has $L$ layers. For layer $l$, we learn a collection of $K$ cross-data shared expert weights $\boldsymbol{\theta}^l = \{\boldsymbol{\theta}_1^l, \boldsymbol{\theta}_2^l, \cdots, \boldsymbol{\theta}_K^l\}$ (fully connected layers) and a unique latent $\mathbf{h}^l \in \mathbb{R}^H$ for each data instance. At inference, the operation at layer $l$ is $\boldsymbol{y}^{l+1} = \sin(\omega_0 \cdot (\bar{\boldsymbol{\theta}}^l \cdot \boldsymbol{y}^l))$, where $\boldsymbol{y}$ represents each layer's output and $\omega_0$ is a constant factor. $\bar{\boldsymbol{\theta}}^l = \sum_{n=1}^K \boldsymbol{\theta}_k^l \cdot \alpha_k^l$ denotes instance-specific parameters at layer $l$, modulated by a *gating vector* $\boldsymbol{\alpha}^l$, which is computed by a gating module $g_{\boldsymbol{\phi}}(\mathbf{h}^l) = \boldsymbol{\alpha}^l = [\alpha_1^l, \alpha_2^l, \cdots, \alpha_K^l]^\top$. Compared with directly learning the gating vectors, $g_{\boldsymbol{\phi}}(\cdot)$ allows for a more compact latent $\mathbf{h}^l$ that benefits distribution learning. By modulating the contribution of experts via latents, each layer gains the flexibility to adapt to individual data samples with a shared basis. As $L$ layers are cascaded to form the final INR, its expressive capacity is significantly enhanced through the integrated contributions across layers.

In Stage-1, we optimize the shared network parameters $\boldsymbol{\theta} = \{\boldsymbol{\theta}^1, \boldsymbol{\theta}^2, \cdots, \boldsymbol{\theta}^L, \boldsymbol{\phi}\}$ and fully characterize each instance-specific INR by layer-wise stacked latents $\mathbf{h} = [\mathbf{h}^1, \mathbf{h}^2, \cdots, \mathbf{h}^L] \in \mathbb{R}^{H \times L}$. This layer-wise structure enables hierarchical modeling of INR parameters, which aligns with data semantic hierarchy, allowing for layer-wise dependency modeling in Stage-2, and controllable data generation.

### 3.2. Stage-1: Learning a Dataset of LoE INRs

Similar to Functa (Dupont et al., 2022a) and mNIF (You et al., 2024), we use meta-learning and auto-decoding to train both the data-specific latents $\mathbf{h}$ and the shared parameters $\boldsymbol{\theta}$ for the LoE INR during Stage-1. For meta-

learning, we adopt an interleaved training procedure inspired by CAVIA (Zintgraf et al., 2019), where the experts and latents are updated alternately in separate training loops. In the inner loop, we fix $\boldsymbol{\theta}$ and adapt the latents $\mathbf{h}$ to data samples. Within each inner loop, $\mathbf{h}$ is first randomly initialized around zero and then updated for a few steps. In the outer loop, $\boldsymbol{\theta}$ is optimized based on the updated $\mathbf{h}$. This ensures each data-specific latent can be effectively learned within a few iterations, encouraging faster convergence and adaptation to new data, which is essential for distribution modeling and generalization in Stage-2. In the case of auto-decoding, we jointly optimize all parameters, maintaining a latent bank for the dataset and updating the sampled batch of latents in each iteration. Unlike meta-learning, auto-decoding does not require second-order derivatives, making it more computationally efficient. Due to this efficiency, we apply auto-decoding specifically for NeRF training.

In both approaches, each data-specific latent $\mathbf{h}$ consists of $L$ components separately modulating $L$ layers in LoE INR. This setup facilitates conditional distribution modeling in Stage-2, as opposed to joint distribution learning like mNIF and Functa, enabling hierarchical and controllable generation—a crucial capacity lacking in prior works.

### 3.3. Stage-2: Conditional Distribution Learning

Given a collection of latents $\mathcal{H} = \{\mathbf{h}_1, \cdots, \mathbf{h}_N | \mathbf{h}_n \in \mathbb{R}^{H \times L}\}$ obtained from $N$ data instances, where $L$ denotes number of layers, and $H$ the dimension of each layer's latent, Stage-2 aims at learning latent distribution $p(\mathbf{h})$. Instead of blindly modeling joint distribution, we reformulate $p(\mathbf{h})$ as:

$$p(\mathbf{h}) = p(\mathbf{h}^1, \mathbf{h}^2, \cdots, \mathbf{h}^L) = p(\mathbf{h}^1) \prod_{l=2}^L p(\mathbf{h}^l | \mathbf{h}^{<l}), \quad (3)$$

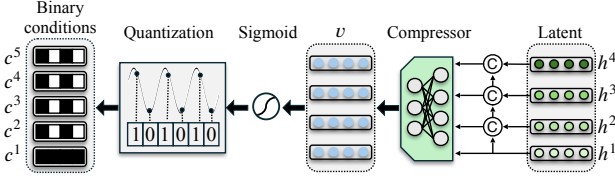

*Figure 3.* Condition formation. Layers of latent **h** are first concatenated and then compressed. The compressed tensors are binarized to generate low-dimensional binary conditions.

where $p(\mathbf{h}^{<l}) = p(\mathbf{h}^1, \cdots, \mathbf{h}^{l-1})$ denotes the joint probability of the first $l-1$ layers. We design a hierarchical conditional diffusion model (HCDM) that learns the conditional dependency $p(\mathbf{h}^l|\mathbf{h}^{<l})$ in Eq. (3). Figure 2 *right* illustrates HCDM with a forward and a backward process.

### 3.3.1. FORWARD PROCESS

We initialize **h** at step 0 as $\mathbf{h}_0 = [\mathbf{h}_0^1, \cdots, \mathbf{h}_0^L]$ with a conditional chain of length $L$. The forward process for each layer $\mathbf{h}^l$ is formulated as:

$$q(\mathbf{h}_{1:T}^l|\mathbf{h}_0^l) := \prod_{t=1}^{T} q(\mathbf{h}_t^l|\mathbf{h}_{t-1}^l), \qquad (4)$$
$$q(\mathbf{h}_t^l|\mathbf{h}_{t-1}^l) := \mathcal{N}(\mathbf{h}_t^l; \sqrt{1-\beta_t}\mathbf{h}_{t-1}^l, \beta_t\mathbf{I}),$$

where $q(\mathbf{h}_t^l|\mathbf{h}_{t-1}^l)$ denotes the posterior distribution of $\mathbf{h}_t^l$ given $\mathbf{h}_{t-1}^l$, $T$ denotes the number of diffusion steps. $\beta_1, \cdots, \beta_T$ denote the variance schedule of added Gaussian noise $\mathcal{N}(\cdot)$. By the forward process, the noise sample $\mathbf{h}_T = [\mathbf{h}_T^1, \cdots, \mathbf{h}_T^L]$ is generated from $\mathbf{h}_0 = [\mathbf{h}_0^1, \cdots, \mathbf{h}_0^L]$.

### 3.3.2. BACKWARD PROCESS

The backward process models the prior distribution as in Eq. (3). To model the hierarchical structure, this process should express the conditional dependency $p(\mathbf{h}^l|\mathbf{h}^{<l})$ for all $L$ components. Therefore, we take $\mathbf{h}^{<l}$ as condition, and generate $\mathbf{h}^l$ for $l = 1, \cdots L$ iteratively. Next, we explain the details of how to process $\mathbf{h}^{<l}$ as condition.

**Condition formation.** To generate $\mathbf{h} = [\mathbf{h}^1, \cdots, \mathbf{h}^L]$, we prepare for each $\mathbf{h}^l$ a condition vector $\mathbf{c}^l$ encapsulating $\mathbf{h}^{<l}$. Since **h** lives in a low-dimensional manifold (e.g., 64), $\mathbf{c}^l$ should contain less information to prevent HCDM from *memorizing* all one-to-one mappings ($\mathbf{h}^{<l} \rightarrow \mathbf{h}^l$), where $p(\mathbf{h})$ inevitably degenerates into $p(\mathbf{h}^1)$. Therefore, we embed the conditions $\mathbf{h}^{<l}$ into a lower-dimensional binary vector. Fig. 3 shows this process with two steps. (1) All $\mathbf{h}^{<l}$ are concatenated and compressed with a compressor **W** to get a compressed tensor $\mathbf{v}^l \in \mathbb{R}^C$: $\mathbf{v}^l = \mathbf{W} \cdot \mathrm{concat}(\mathbf{h}^1, \cdots, \mathbf{h}^L), \mathbf{W} \in \mathbb{R}^{C \times HL}, \mathbf{h}^j \in \mathbb{R}^H$, with $C$ being the compressed dimension. To match **W**'s dimension

with concatenated tensors, we set its unused portion to zero. (2) Given compressed $\mathbf{v}^l$, we obtain a binary condition $\mathbf{c}^l$: $\mathbf{c}^l = \mathcal{Q}(\sigma(\mathbf{v}^l))$, where $\mathcal{Q}(\cdot)$ denotes binarization operation, $\sigma$ denotes Sigmoid. We set $\mathbf{c}^1$ as zero tensor since $\mathbf{h}^1$ has no condition. Now we get the condition $\mathbf{c}^1, \cdots, \mathbf{c}^L$.

**Hierarchical generation.** With condition $\mathbf{c} = [\mathbf{c}^1, \cdots, \mathbf{c}^L]$, noise sample $\mathbf{h}_T = [\mathbf{h}_T^1, \cdots, \mathbf{h}_T^L]$, time step $t$, we are ready to generate a complete latent $\mathbf{h}_0$. To generate a component $\mathbf{h}_0^l$, the backward process is formulated as: $p(\mathbf{h}_{0:T}^l|\mathbf{c}^l) = p(\mathbf{h}_T^l) \prod_{t=1}^{T} p(\mathbf{h}_{t-1}^l|\mathbf{h}_t^l, \mathbf{c}^l)$. Note that a complete backward process, generating a sample from $p(\mathbf{h}_0^l|\mathbf{c}^l)$, is exactly the implementation of $p(\mathbf{h}^l|\mathbf{h}^{<l})$ in Eq. (3), where $\mathbf{c}^l$ corresponds to $\mathbf{h}^{<l}$. We adopt a UNet (Ronneberger et al., 2015) $\mu_{\boldsymbol{\theta}}$ as in Song et al. (2021); Ho et al. (2020) to obtain $p(\mathbf{h}_{t-1}^l|\mathbf{h}_t^l, \mathbf{c}^l) = \mathcal{N}(\mathbf{h}_{t-1}^l : \boldsymbol{\epsilon}_{\boldsymbol{\theta}}(\mathbf{h}_t^l, t, \mathbf{c}^l), \boldsymbol{\Sigma}(t))$, where

$$\boldsymbol{\epsilon}_{\boldsymbol{\theta}}(\mathbf{h}_t^l, t, \mathbf{c}^l) = \frac{\sqrt{a_t(1-\bar{a}_{t-1})}\mathbf{h}_t^l + \sqrt{\bar{a}_{t-1}(1-a_t)}\mu_{\boldsymbol{\theta}}(\mathbf{h}_t^l, t, \mathbf{c}^l)}{1-\bar{a}_t}$$
$$a_t = 1 - \beta_t, \ \bar{a}_t = \prod_{i=1}^{t} a_i, \ \boldsymbol{\Sigma}(t) = \frac{(1-a_t)(1-\bar{a}_{t-1})}{1-\bar{a}_t}\mathbf{I}.$$

$$(5)$$

Each backward process starts from a noise $\mathbf{h}_T^l$ and generates a latent component $\mathbf{h}_0^l$ with condition $\mathbf{c}^l$. By iteratively sampling from $p(\mathbf{h}^1)$ and $p(\mathbf{h}^l|\mathbf{h}^{<l})$ with $l = 2, \cdots L$, a complete latent $\mathbf{h}_0 = [\mathbf{h}_0^1, \cdots, \mathbf{h}_0^L]$ is generated. The final objective is: $\mathcal{L}_{hcdm} = \mathbb{E}_{\mathcal{H},t,l}[\lambda||\epsilon - \epsilon_{\boldsymbol{\theta}}(\mathbf{h}_t^l, t, \mathbf{c}^l)||^2]$, where $\mathcal{H}$ denotes all latents, $\epsilon$ denotes Gaussian sample from $\mathcal{N}(\mathbf{0}, \mathbf{I})$, and $\lambda$ is a constant coefficient. For inference, we first generate $\mathbf{h}^1$ from Gaussian noise, then perform a chain of conditional sampling from $p(\mathbf{h}^l|\mathbf{h}^{<l})$ until we get the complete latent **h** used by the LoE to generate new data.

## 4. Experiments

In this section, we first introduce the datasets and evaluation criteria. We then show CHINR's controllability of the generation process across various modalities. A quantitative evaluation is provided to highlight its hierarchical control over semantic attributes. Next, we evaluate CHINR's reconstruction and generation quality. Furthermore, we analyze the latent space, showing how data semantics are embedded within the INR's weight space, granting it a *compositional* property across layers. Finally, we present ablation studies highlighting the importance of conditional dependency modeling and the functionality of binary conditions.

In our experiments, both Stage-1 and Stage-2 are trained and evaluated on the CelebA-HQ $64^2$ (Karras et al., 2018), ShapeNet $64^3$ (Chang et al., 2015), SRN-Cars (Sitzmann et al., 2019), and AMASS (Mahmood et al., 2019) datasets. All experiments are implemented in Pytorch and run on a single Nvidia RTX3090 GPU. For evaluation metrics, we use peak signal-to-noise ratio (PSNR), structural sim-

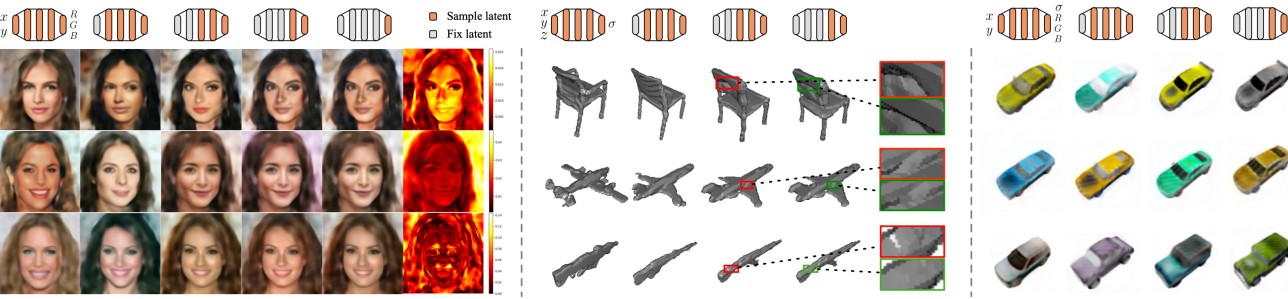

*Figure 4.* Controllable hierarchical generation by progressively fixing the layer-wise latent on three modalities.

ilarity (SSIM) ([Wang et al., 2004](#)), and accuracy to assess Stage-1 reconstruction, and Fréchet inception distance (FID) ([Heusel et al., 2017](#)), and coverage to evaluate Stage-2 generation performance. Further details about implementation can be found in the Appendix.

### 4.1. Hierarchical Controllable Generation

In this section, we show hierarchical controllable generation enabled by conditional sampling of HCDM, a crucial capability missing in existing works. To sample a latent $\mathbf{h} = [\mathbf{h}^1, \cdots, \mathbf{h}^5]$, we begin by sampling $\mathbf{h}^1$ with HCDM taking a Gaussian noise as input. Next, we sample $\mathbf{h}^2$, conditioned on the binary vector generated by $\mathbf{h}^1$. This process continues layer by layer until $\mathbf{h}^5$ is generated, forming a complete latent $\mathbf{h}$. This layer-wise latent then modulates the LoE to produce images, voxels, or NeRF renderings.

**Qualitative results.** Fig. 4 shows hierarchical control on different modalities. For each, we show three samples generated through the full chain of conditional sampling $(\mathbf{h}^1 \cdots \mathbf{h}^5)$ in *column 1*. In *column 2*, we fix their $\mathbf{h}^1$ from the first column and sample the remaining $(\mathbf{h}^2 \cdots \mathbf{h}^5)$. In *column 3*, we fix both $\mathbf{h}^1$ and $\mathbf{h}^2$ from the second column and conditionally sample the rest. This progressive fixation allows us to control the finer details of the generated output.

In Fig. 4, each run starts from a different $\mathbf{h}^1$, resulting in highly different semantics (*column 1* in each modality). For face images, when $\mathbf{h}^1$ is fixed, generations exhibit similar overall contours but different facial features and hairstyles (*column 2*). Fixing $\mathbf{h}^1$ and $\mathbf{h}^2$ causes variations in facial details, like eye shape and hair color (*column 3*). Fixing first three layers (*column 4*) results in changes limited to skin tone. Lastly, $\mathbf{h}^5$ affects global properties like foreground/background color, shown by heat maps. For voxels, hierarchy control is manifested in category (e.g. chair and plane), object parts (e.g. armrest and cushion), and finer details (e.g. textures). For NeRF, hierarchy occurs at car type (e.g. SUV), shape (e.g. boxy), parts (e.g. spoiler), etc. Additional examples are provided in the Appendix.

**Quantitative evaluation of hierarchical control.** We fur-

*Table 1.* Quantitative evaluation of hierarchical control on CelebA-HQ (*top*), ShapeNet (*middle*), and SRN-Cars (*bottom*).

| Attributes | Random | $L_1$ | $L_2$ | $L_3$ | $L_4$ |
|---|---|---|---|---|---|
| Oval face | 30.3% | **73.2%** | 85.7% | 90.2% | 94.1% |
| Blonde hair | 39.6% | 40.5% | **85.8%** | 91.4% | 96.8% |
| Smiling | 30.8% | 31.2% | 33.6% | **83.0%** | 92.5% |
| Red lips | 10.5% | 11.3% | 13.7% | 12.5% | **95.3%** |
| Chair | 12.8% | **93.6%** | 98.9% | 100.0% | 100.0% |
| w/ Arms (Chair) | 26.3% | 29.3% | **78.8%** | 97.0% | 100.0% |
| Cushioned (Chair) | 19.2% | 21.2% | 32.3% | **87.9%** | 100.0% |
| SUV | 19.4% | **76.9%** | 94.8% | 97.8% | 100.0% |
| Boxy | 32.8% | 46.3% | **88.8%** | 96.3% | 97.0% |
| w/ Spoiler | 14.9% | 19.4% | 24.6% | **68.7%** | 89.6% |

ther quantitatively evaluate controllability in Table 1. We select representative attributes and start by randomly generating (w/o control) samples, then compute the ratio of samples where each attribute is present (column *Random*). For each attribute, we take the positive samples and progressively fix the latents of initial layers while sampling the rest (Columns $L_{1-4}$). At each step, we compute the positive ratio of the newly generated samples to verify whether an attribute is preserved. This experiment assesses the model's ability to maintain control over fixed attributes while varying others. The results show the effectiveness of coarse-to-fine, layer-specific control enabled by the latent hierarchy. Bold numbers denote a significant increase in preservation rate compared to the previous layer, showing that the corresponding attributes are effectively controlled at that layer.

### 4.2. Reconstruction & Generation Metrics

Table 2 presents reconstruction and generation results on different modalities. Our model outperforms existing methods on most datasets. It achieves the highest reconstruction PSNR, thanks to the expanded representation capacity of LoE and layer-wise latent learning. It also shows superior generation performance, highlighting the effectiveness of hierarchical conditional modeling in capturing diverse data semantics. On CelebA-HQ, its FID score is slightly behind

*Table 2.* Quantitative results on different datasets.

| CelebA-HQ | Reconstruction | | Generation |
|---|---|---|---|
| | PSNR↑ | SSIM↑ | FID↓ |
| Functa (Dupont et al., 2022a) | 26.2±0.3 | 0.795±0.015 | 41.0±0.2 |
| GEM (Du et al., 2021) | 26.5±0.4 | 0.814±0.018 | 30.8±0.3 |
| GASP (Dupont et al., 2022c) | 31.6±0.8 | 0.902±0.021 | 13.6±0.3 |
| mNIF (You et al., 2024) | 34.5±0.2 | 0.957±0.005 | **13.2±0.1** |
| CHINR | **34.9±0.3** | **0.964±0.006** | 13.4±0.1 |
| **ShapeNet** | PSNR↑ | Accuracy↑ | Coverage↑ |
| Functa (Dupont et al., 2022a) | 22.1±0.3 | 0.983±0.005 | 0.437±0.005 |
| GEM (Du et al., 2021) | 21.4±0.4 | 0.977±0.007 | 0.408±0.003 |
| GASP (Dupont et al., 2022c) | 16.7±0.8 | 0.928±0.011 | 0.343±0.009 |
| mNIF (You et al., 2024) | 21.4±0.3 | 0.975±0.008 | 0.435±0.003 |
| CHINR | **22.3±0.2** | **0.988±0.005** | **0.441±0.002** |
| **SRN-Cars** | PSNR↑ | SSIM↑ | FID↓ |
| Functa (Dupont et al., 2022a) | 24.3±0.2 | 0.738±0.009 | 80.1±0.2 |
| mNIF (You et al., 2024) | 26.0±0.3 | 0.763±0.013 | 79.3±0.3 |
| CHINR | **26.3±0.2** | **0.780±0.011** | **77.8±0.2** |

mNIF. However, we observe that mNIF tends to "memorize" the training set. To verify this, we generate $1,000$ samples and compute the average of their minimum L2 distances to training images. A lower value indicates a higher degree of memorization. mNIF obtains a value of 6.243 whereas CHINR obtains 15.971, indicating that our model generalizes better by generating new images that differ more noticeably from training data. Examples of retrieval results are provided in the Appendix.

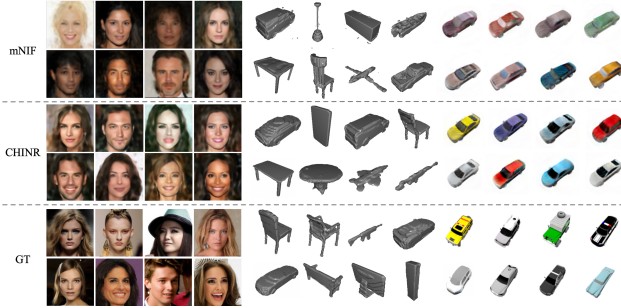

*Figure 5.* Uncurated generations for CelebA-HQ (*left*), ShapeNet (*middle*), and SRN-Cars (*right*) datasets.

Fig. 5 displays uncurated samples generated by our model compared to mNIF. With HCDM, our model generates high-quality samples with rich details. More results including the AMASS dataset are provided in the Appendix.

### 4.3. Analysis

We uncover the success of hierarchical controllable generation. We first show that each layer modulates disentangled semantics, giving our LoE model a compositional property. Moreover, layer-wise conditional dependency modeling successfully captures the hierarchy of data semantics, ensuring

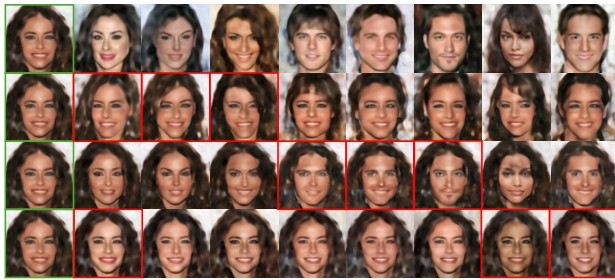

*Figure 6.* Latent composition. *top*: 9 randomly generated images. *row 2-4*: replace $\mathbf{h}^2$ to $\mathbf{h}^4$ of the first sample (green boxes) with *top* images. Red boxes highlight representative examples.

the composed semantics are compatible across layers.

#### 4.3.1. LATENT COMPOSITION

We explore CHINR's controllability through latent composition. We find that the learned latents are layer-wise compositional. Given two latents $\mathbf{h}_1 = [\mathbf{h}_1^1, \cdots, \mathbf{h}_1^5]$, $\mathbf{h}_2 = [\mathbf{h}_2^1, \cdots, \mathbf{h}_2^5]$, exchanging a specific part, e.g. $\mathbf{h}_1^2$ and $\mathbf{h}_2^2$, results in the corresponding semantic changes in the generated content. As shown in Fig. 6, we randomly sample 9 latents from HCDM, trained on CelebA-HQ, and render the images in the first row. These faces exhibit diverse characteristics including expressions, hairstyles, facial orientations, skin tones, etc. In the second row, we replace $\mathbf{h}^2$ of the first sample with that of the other 8 samples, while keeping the rest latents unchanged. The facial orientation and hairstyles change accordingly, while facial features remain the same. This indicates that the second layer encodes these specific semantics disentangled from others, laying the foundation for controllable hierarchical generation. In the third row, when $\mathbf{h}^3$ is replaced, we observe that only facial features are swapped, while other characteristics such as orientation remain unchanged. In the last row, only skin tone changes. More examples can be found in the Appendix.

It's noted that latent composition can disrupt the conditional chain, leading to incompatible latents and thus image artifacts. Nevertheless, this experiment illustrates how and why our method works. It shows that image semantics can be embedded and disentangled in parameter space, offering a new perspective on image generation.

#### 4.3.2. LAYER-WISE HIERARCHY ANALYSIS

We visualize hierarchical dependencies of latents to see how data semantics are encoded in INR's weight space. Fig. 7 shows conditional distributions of latents from adjacent layers, trained on CelebA-HQ. Specifically, we apply PCA to Stage-1 latents and plot their distributions in grayscale (e.g., Fig. 7a shows distributions for layers 1 and 2). We then run the generation process five times to obtain five latents at

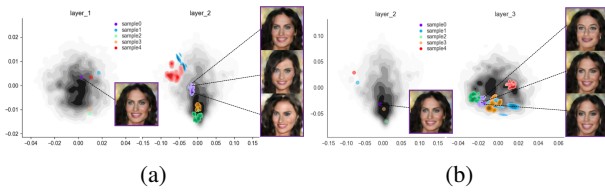

(a)                                      (b)

*Figure 7.* Visualization of layer-wise hierarchical dependencies. Gray regions show the distribution of latents from Stage-1, while colored regions represent the sampled latents from Stage-2.

each layer, as depicted in color Figs. 7a and 7b *left*. Based on HCDM, we plot the resulting conditional distributions of latents at subsequent layers, shown by colored regions on the *right*. Additionally, we show generated images corresponding to different samples. We can see clear patterns of a hierarchical structure where semantics vary at different granularities. For example, layer 1 determines the overall contours of faces. When layer 2 is determined, variations in layer 3 affect facial expressions without changing orientations. Furthermore, the latent sampling space at each layer is constrained by its preceding layer, ensuring compatibility between layers when representing data. Therefore, the generated data semantics are hierarchically controllable.

### 4.4. Ablation Studies

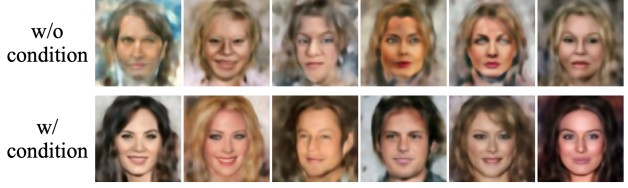

*Figure 8.* Ablations on conditional modeling.

*Table 3.* Ablation on number of layers in conditional chain.

| Layers in chain | 1, 2, 3, 4, 5 | 1, 2, 3, 4 | 1, 2, 3 | 1, 2 | None |
|---|---|---|---|---|---|
| Independent layers | None | 5 | 4, 5 | 3, 4, 5 | 1, 2, 3, 4, 5 |
| FID | 13.4 | 13.6 | 15.5 | 52.8 | 112.7 |

**Ablation on condition modeling.** To show the importance of conditional dependency modeling, we train an unconditional diffusion model that directly maps noise to layer-wise latents in Stage-2. We then sample $p(\mathbf{h}^l)$ independently for $l = 1, \cdots, L$ to generate full latents. The resulting images for CelebA-HQ are shown in Fig. 8 *top*, while images generated with conditional modeling are shown in *bottom* for comparison. Although human faces are recognizable in *top*, the noticeable artifacts highlight that independently sampled layer-wise latents cannot ensure consistent semantic composition across layers. In contrast, conditional modeling successfully achieves this compatibility. In Table 3,

we evaluate the importance of conditional dependency by progressive ablation where excluded layers are trained with unconditional diffusion models. Results show that strengthening conditional modeling improves generation quality, emphasizing its necessity for high-quality generations.

*Table 4.* Ablation on different binary condition lengths (8, 12, 15, and 20) when training Stage-2 on CelebA-HQ.

| Model | $\mathbf{std}_1$ | $\mathbf{std}_{12}$ | $\mathbf{std}_{23}$ | $\mathbf{std}_{34}$ | $\mathbf{std}_{45}$ |
|---|---|---|---|---|---|
| $HCDM_8$ | 0.7766 | 0.9153 | 0.9363 | 0.8772 | 0.8631 |
| $HCDM_{12}$ | 0.7766 | 0.9092 | 0.9292 | 0.8663 | 0.8550 |
| $HCDM_{15}$ | 0.7766 | 0.5344 | 0.5478 | 0.3257 | 0.2570 |
| $HCDM_{20}$ | 0.7766 | 0.1032 | 0.1121 | 0.0853 | 0.0766 |

**Ablation on binary condition.** We show that the length of binary conditions impacts the effectiveness of learning conditional dependencies, as in Table 4. We set binary lengths to 8, 12, 15, 20 and train HCDM on CelebA-HQ. Initially, we sample $5,000$ latents and compute the standard deviation ($\mathbf{std}_1$) of $\mathbf{h}^1$. Since $\mathbf{h}^1$ is sampled w/o condition, it shows high values irrelevant to binary lengths. Then, we select 10 random samples from $\mathbf{h}^1$ as conditions and get $5,000$ samples for $\mathbf{h}^2$. The standard deviation ($\mathbf{std}_{12}$) decreases as the binary condition length increases since longer conditions contain more information from preceding layers. Once it reaches a certain threshold, the standard deviation approaches zero, turning the conditional chain into direct one-to-one mapping, thus diminishing controllability. However, if the length is too small, e.g. 0, all parts will be independent hence losing conditional dependency. Therefore, we empirically set the length as 12. We repeat this procedure for all layers and observe similar results.

## 5. Conclusion

We proposed the Controllable Hierarchical Implicit Neural Representation (CHINR) framework, addressing the limitations of existing generative INRs which ignore the hierarchy in parameters and data semantics and lack control. By structuring INR as a Layers-of-Experts network and leveraging a hierarchical conditional diffusion model, our approach captures conditional dependencies across layers, enabling hierarchical controllable data generation.

One limitation is scalability to larger datasets, as the shared network may struggle to capture complex and diverse data patterns. A possible solution is to incorporate local information using patch-wise modulation (Mehta et al., 2021; Bauer et al., 2023) or INRs with localized nonlinearity (e.g., WIRE (Saragadam et al., 2023)). Future directions include exploring sparse gatings, as in Mixture of Experts (Wang et al., 2022), to promote expert diversity and specialization. Additionally, learning layer-wise semantic hierarchy in

Stage-1 can be guided through predefined attributes or deep clustering. This would allow the model to develop more interpretable and distinct semantics across layers, improving control over fine-grained details and desired characteristics.

## Impact Statement

This paper advances generative INRs by introducing a novel approach for controllable data generation. Our experiments include datasets that contain human-related data (e.g., facial images), but we do not foresee any immediate societal concerns requiring specific attention. We encourage responsible use of our method in applications involving human data.

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

# A. Details on Experimental Setup

**Implementation details.** The LoE structure can be configured with the number of layers $L$, the number of experts at each layer $K$, the channel dimension of each expert $C$, and the dimension of the latent at each layer $H$, denoted as a tuple $(L, K, C, H)$. We train LoEs of $(5, 128, 256, 128)$, $(5, 256, 64, 256)$, $(6, 256, 64, 64)$, and $(5, 64, 64, 64)$ in CelebA-HQ (Karras et al., 2018), ShapeNet (Chang et al., 2015), SRN-Cars (Sitzmann et al., 2019), and AMASS (Mahmood et al., 2019) datasets, respectively. We follow mNIF (You et al., 2024) on the data processing protocols for CelebA-HQ, ShapeNet, and SRN-Cars datasets. Details about the AMASS dataset are provided in Appendix B.3.

**Training details.** In Stage-1, we train LoEs via meta-learning on CelebA-HQ, ShapeNet, and AMASS, and with auto-decoding on SRN-Cars. We use a batch size of 32, an outer learning rate of $1e-4$, an inner learning rate of 1 with 3 steps, and train the LoE for 800 epochs in the meta-learning setting. For auto-decoding experiments on SRN-Cars, we use a batch size of 8, a learning rate of $1e-4$, and train the LoE for 1000 epochs. In both settings, we use the AdamW (Loshchilov, 2017) optimizer without weight decay. In Stage-2, we set the training batch size to be 32, learning rate $1e-4$, and cosine scheduler with minimum learning rate 0.0. We train the HCDM for 1000 epochs with the AdamW optimizer.

# B. Additional Experimental Results

## B.1. Generalizability Analysis Through Retrieval

Generated                          Retrieved from training set

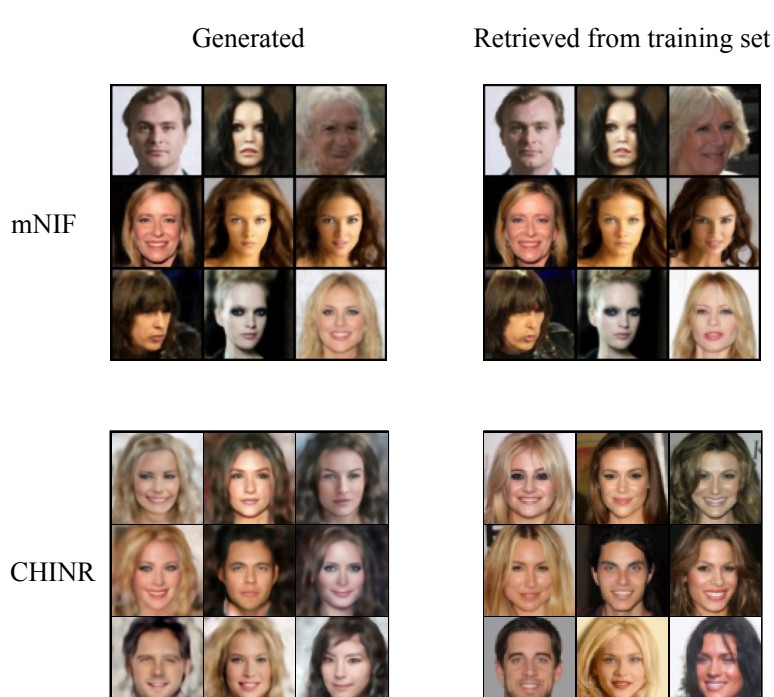

*Figure 9.* Retrieval on CelebA-HQ: mNIF retrieves images closely resembling those from the training set, while CHINR demonstrates better generalization by producing distinct new images.

We use retrieval to compare the generalizability of CHINR and mNIF on the CelebA-HQ dataset. Specifically, we generate samples and retrieve the closest images from the training set. As shown in Fig. 9, mNIF generates samples that are very similar to the training images, suggesting a higher chance of "memorization". In contrast, CHINR demonstrates better generalization by producing "new" samples that differ more noticeably from the training data.

## B.2. More Generated Samples

Figure 10 shows more generated samples on CelebA-Net, ShapeNet, and SRN-Cars datasets.

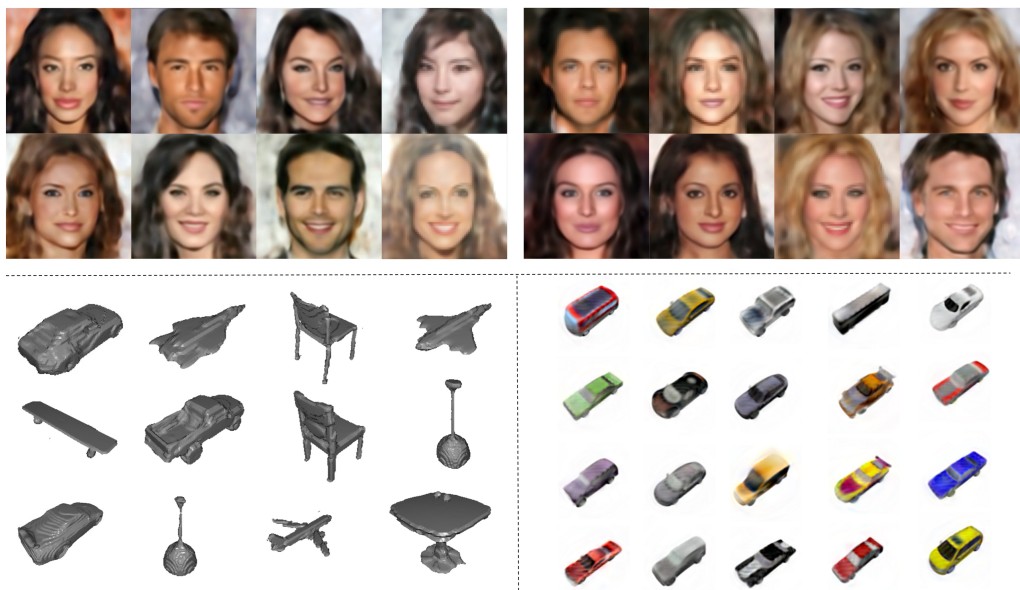

*Figure 10.* More generated samples of CelebA-HQ, ShapeNet, and SRN-Cars data.

*Table 5.* Quantitative results on AMASS.

| Model | MSE↓ |
|---|---|
| mNIF (You et al., 2024) | 0.015 |
| CHINR | 0.011 |

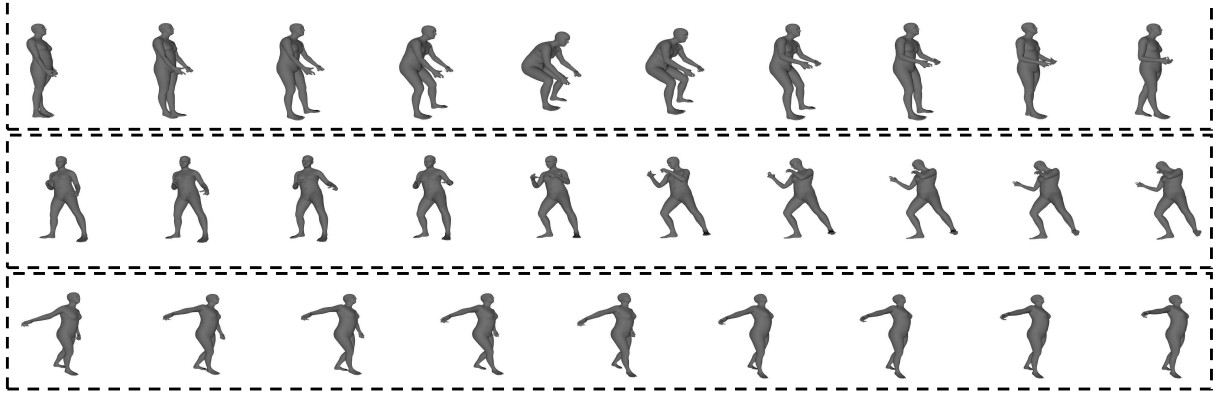

*Figure 11.* Generated motions with HCDM. each row denotes one sampled data.

### B.3. AMASS Experiments

We apply our proposed CHINR model to the AMASS dataset of 3D human motions. For each motion sequence, we use 200 frames, with each frame represented by 165 values corresponding to the locations and rotations of body joints. As a result, each data instance is formatted as a grid with size $200 \times 165$. In Stage-1, the LoE is employed to fit the motion instances. In Stage-2, we set the binary lengths to 8 to avoid memorizing conditions.

**Reconstruction and generation results.** The reconstruction performance is shown in Table 5. The randomly generated motions are shown in Fig. 11.

**Semantic-level Interpolation.** Since the LoE successfully learns the consistent latent space, we can perform semantic-level

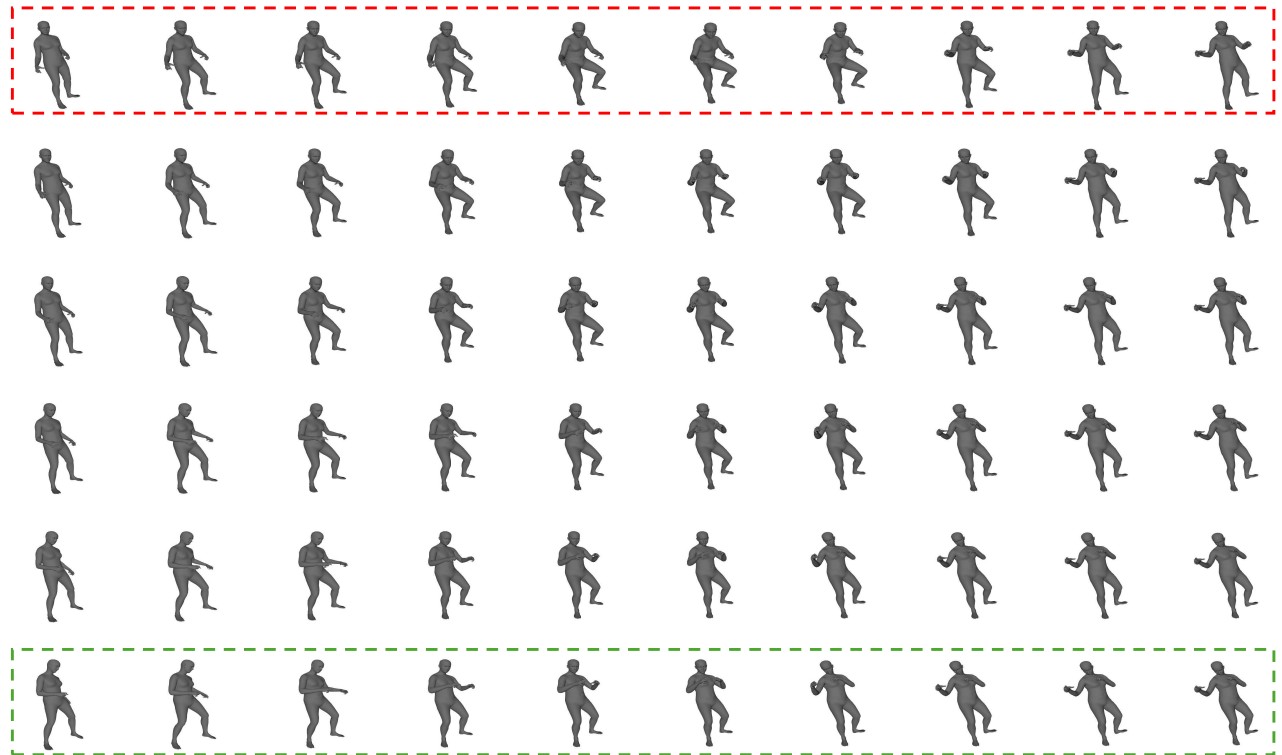

*Figure 12.* Semantic interpolation for AMASS data. Anchor sequential motions (indicated by the red and green dashed boxes) are first fitted with LoE to obtain latents. Then semantic-level interpolation is performed by interpolating the latents. The red dashed box denotes the start motion, and the green dashed box denotes the end motion.

interpolation for motions. As shown in Fig. 12, given two fitted sequential motions with LoE, each corresponds to a latent, we can interpolate the latent from the start motion (indicated by the red dashed box) to the end motion (indicated by the green dashed box) linearly with ratio $[0.2, 0.4, 0.6, 0.8]$. We can see that the interpolated motions change smoothly from the start to the end. Semantic-level interpolation can be useful in the gaming industry, and 3D-digital content generation.

**Temporal-level interpolation.** Since the INR can generate data instances in any resolution, we can easily enlarge the input coordinates' resolution in the time dimension to achieve temporal-level interpolation. We set the length of the time dimension to be 200 and 400, then get motions with LoE. The interpolated results are submitted as videos named "motion_short.mp4" and "motion_long.mp4".

### B.4. Hierarchical Controllable Generation

More examples of hierarchical controllable data generation are presented in Fig. 13.

### B.5. Latent-based Retrieval

We show an application of data retrieval by latents, since they already embed rich semantic meanings. We first obtain the latents for the target data by fitting it to the LoE through a few gradient steps. Once the latents are optimized, they can be used to retrieve similar data by comparing their latent representations to the searched set, allowing us to search for semantically similar examples within the latent space. Figure 14 shows this process by using images from the test-split of CelebA-HQ as the targets, and train-split images as the searched set. We demonstrate two approaches for retrieval: (1) using the flattened $\mathbf{h}$ for all layers, and (2) layer-wise retrieval using each layer's latent $\mathbf{h}^l$. As shown in Fig. 14, retrieval by the flattened $\mathbf{h}$ will retrieve samples that are broadly similar, while layer-wise retrieval retrieves samples with specific semantic similarities. For example, $\mathbf{h}^2$ retrieves faces with similar orientations, while $\mathbf{h}^3$ retrieves faces with similar facial features such as eye shape and expressions.

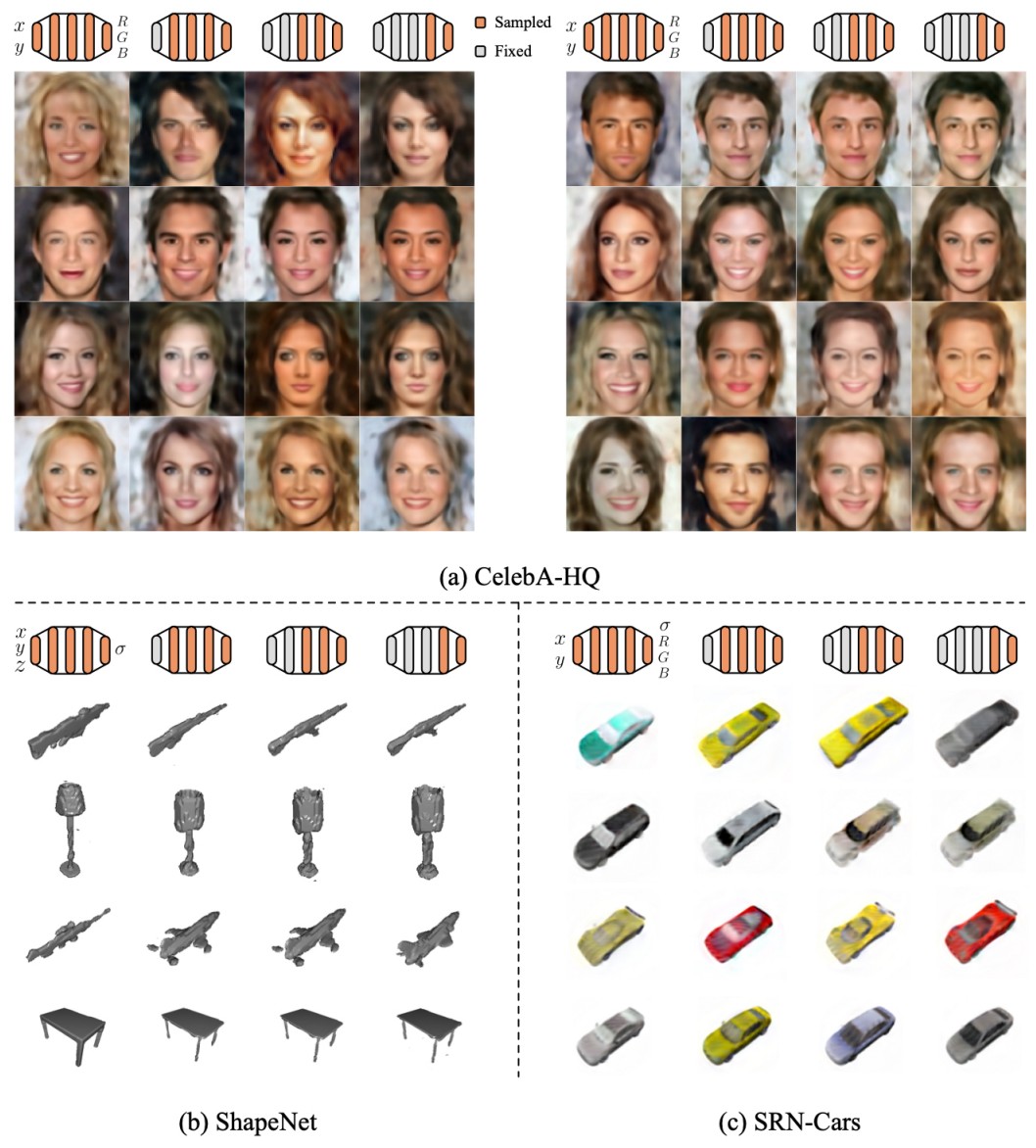

(a) CelebA-HQ

(b) ShapeNet

(c) SRN-Cars

*Figure 13.* More examples of hierarchical controllable generation on CelebA-HQ, ShapeNet, and SRN-Cars data.

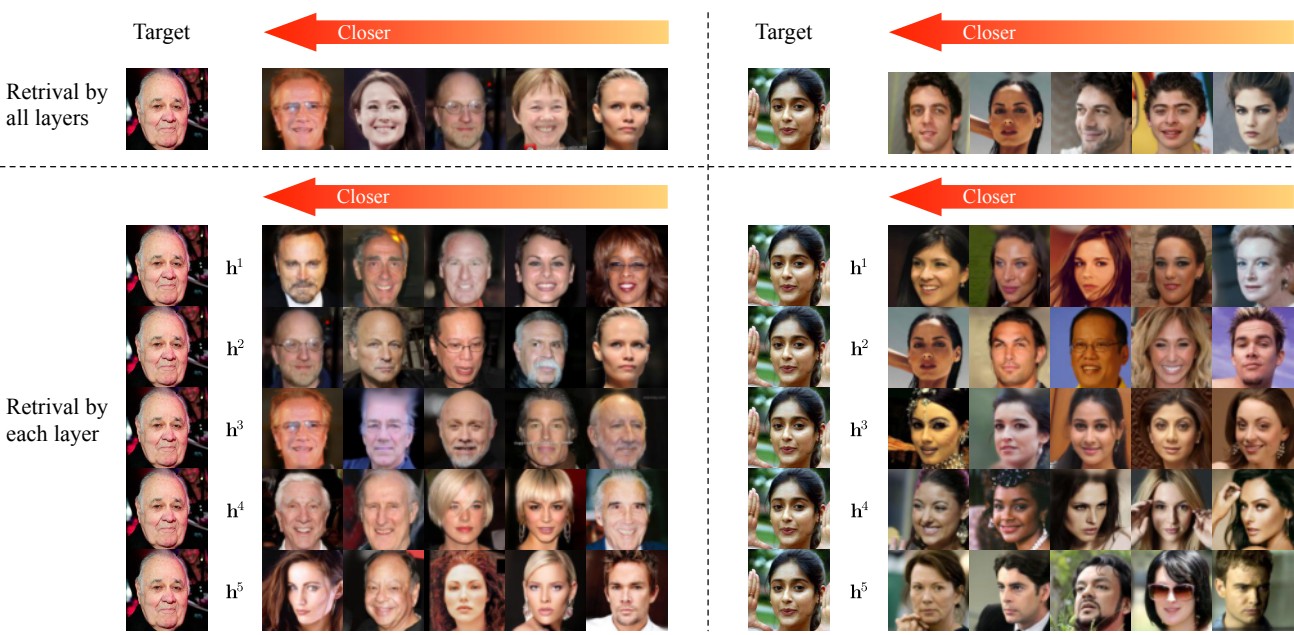

*Figure 14.* Latent-based retrieval via two approaches: retrieval by all layers and retrieval by each layer.

## C. Analysis

In this section, we provide more analysis of the latent space and the functionalities of binary conditions.

### C.1. Latent Space Analysis

Here, we analyze the latent space further, focusing on its interpolation capabilities and providing more results of hierarchy analysis.

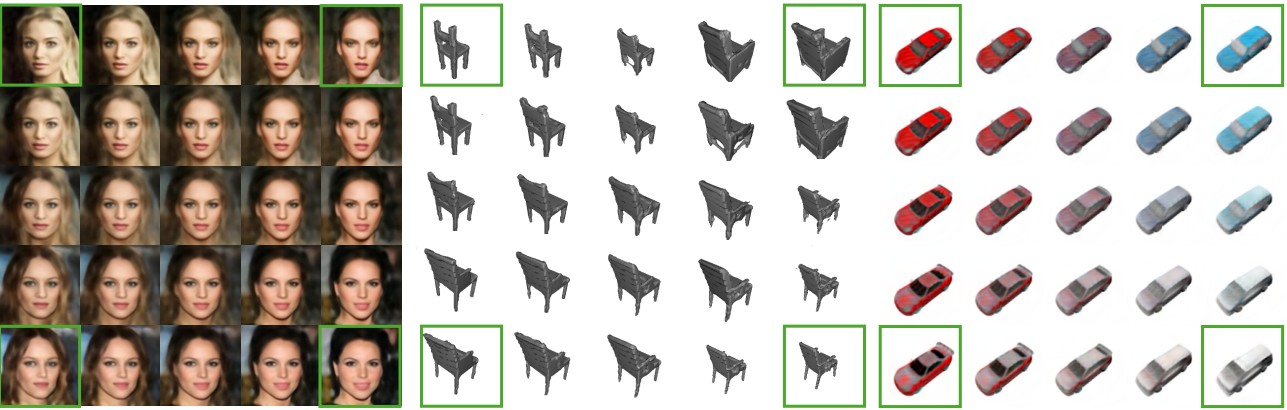

*Figure 15.* Latent space interpolation is performed for LoE, with four corner points representing the anchor examples rendered in stage 1. The intermediary points are generated through the bilinear interpolation of the latents associated with these four anchors. The interpolation is evaluated on datasets CelebA-HQ, ShapeNet, and SRN-Cars.

#### C.1.1. LATENT INTERPOLATION

To illustrate that our model learns a consistent and metric latent space, following definitions in Du et al. (2021), we perform latent space interpolation in two ways: complete interpolation, and layer-wise interpolation.

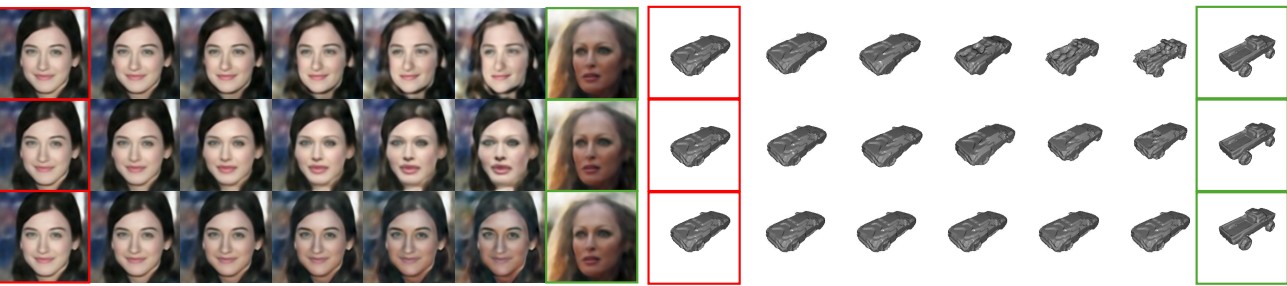

*Figure 16.* Layerwise interpolation. The red boxes denote the start and the green boxes denote the end. For the CelebA-HQ, the layers $2 \to 4$ are interpolated respectively while other layers are fixed. For the ShapeNet, the layers $1 \to 3$ are interpolated respectively.

**Complete Interpolation** is shown in Fig. 15. Four corners present the signals with latent generated from Stage-1. The intermediary signals are bilinearly interpolated from four corners in latent space. The results demonstrate that the learned latent is metric and consistent with human perception.

**Layer-wise Interpolation.** Since our LoE embeds semantics hierarchically in different parts of the latent, we can interpolate each part to control specific semantics. As shown in Fig. 16, we interpolate the second, third, and fourth parts of the latent associated with red-boxed signals, with the corresponding parts of the right side latent. For CelebA-HQ samples, we find that the facial orientation, facial features, and skin tone can be interpolated independently. This demonstrates that each part of the latent also constructs a metric and consistent manifold.

### C.1.2. LAYER-WISE HIERARCHY ANALYSIS

**Layer-wise correlation analysis.** We perform layer-wise correlation analysis on latents to show the necessity of conditional dependency modeling. We compute cross-layer correlation between latents using Singular Vector Canonical Correlation Analysis (SVCCA) (Raghu et al., 2017). Fig. 17a displays the pairwise correlations between $\mathbf{h}$ across layers, trained on Celeba-HQ in Stage-1, showing the non-negligible correlation between layers. Similar findings in other datasets are shown in Figs. 17b and 17c. This underscores the importance of modeling conditional distributions $p(\mathbf{h}^l|\mathbf{h}^{<l})$, rather than independent marginal distributions $p(\mathbf{h}^l)$ in Stage-2.

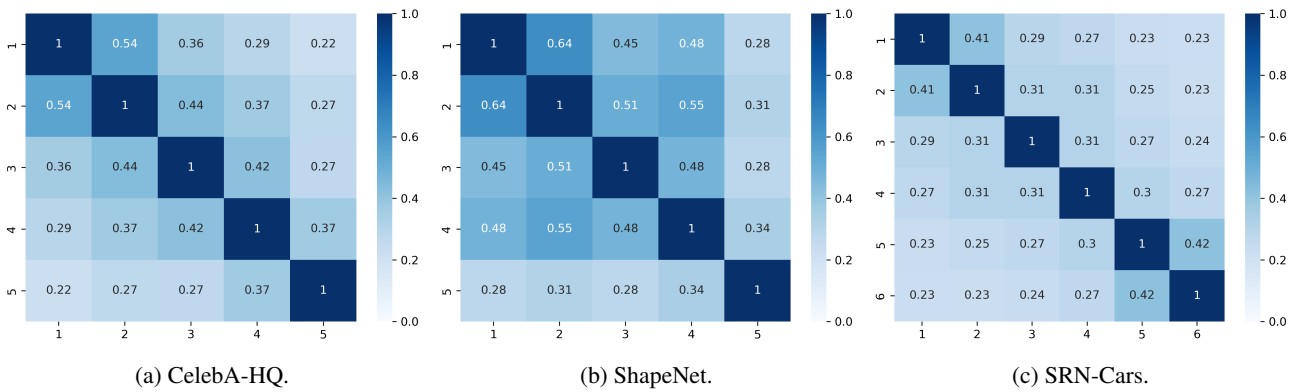

(a) CelebA-HQ.     (b) ShapeNet.     (c) SRN-Cars.

*Figure 17.* Correlation between the learned latents across layers, trained on CelebA-HQ (Karras et al., 2018) ShapeNet (Chang et al., 2015) and SRN-Cars (Sitzmann et al., 2019). The non-negligible correlation between adjacent layers (e.g., $\mathbf{h}^1$ and $\mathbf{h}^2$) reveals the necessity of conditional distribution learning.

**Layer-wise dependency visualization.** We provide more layer-wise dependency visualizations in Fig. 18. This highlights the learned layer-wise dependencies in alignment with data semantic hierarchies.

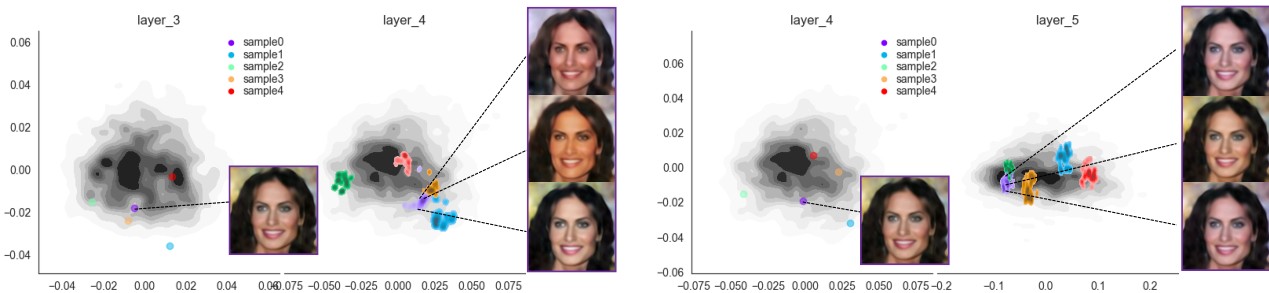

*Figure 18.* Visualization of conditional distributions across layers $3, 4, 5$. The gray regions present the distribution of latents from Stage-1, while the colored regions represent the sampled latents from Stage-2.

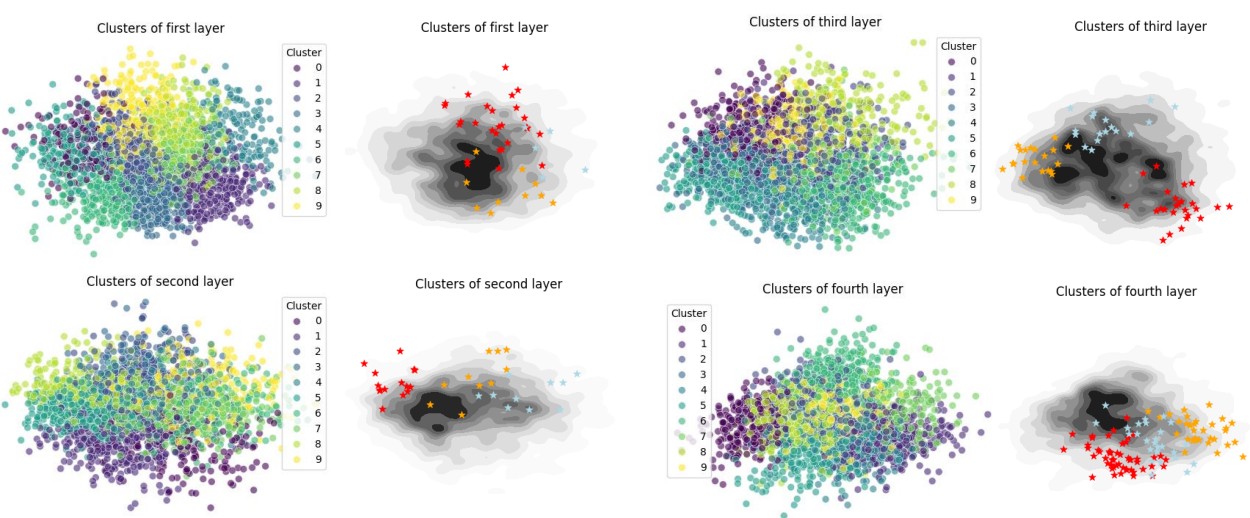

*Figure 19.* Clusters of each part of latent and binary conditions. The dotted plot presents clusters of each part of latents trained on CelebA-HQ. The gray distribution plot presents the distribution of each part of latents, and starred scatter plot presents clusters of latents with similar binary conditions.

## C.2. Binary condition Analysis

We analyze the clustering of latents and binary conditions on CelebA-HQ dataset, as shown in Fig. 19. Firstly, we use the KMeans algorithm to get 10 clusters of latents, shown as the dots in the figure. Then we select three anchor latents, generate three binary conditions with HCDM, and search the nearest binary-corresponded latents. The nearest neighbors are represented by the stars. We can observe that the binary conditions embed the latents' information and form a consistent binary condition space. This binary condition space corresponds to the latent space.

