# OpenReview forum: "Controllable Data Generation with Hierarchical Neural Representations"
_ICML.cc/2025/Conference — ICML 2025 poster_

### Official Review · Reviewer_ZbmH · 2025-03-12

**Overall Recommendation:** 3

**Summary:**

This paper proposes a hierarchical Implicit Neural Representations (INR) framework that aims to provide better control over hierarchical representations during the generation process. In the first stage of the framework, a Layer-of-Experts (LoE) model is trained, and a latent variable is learned for each layer. In the second stage, the hierarchical relationship between different layers is modeled by a conditional diffusion process. Experiments are conducted on several image datasets, and evaluations and comparisons are made with baseline methods.

**Claims And Evidence:**

I found two major issues with the claims and evidence.

First, the paper claims that the proposed framework achieves hierarchical controllable generation. However, this claim remains questionable to me, given the qualitative results shown in Figure 4. While it is clear that the first layer has significant effects on the generation by controlling the fixed “type” of the data, such as the general appearance of a face, the object category in 3D shapes, and vehicle types, the effects of the rest of the layers seem very weak without any clear patterns. For example, which layer or layers control the color of the vehicle, as an important semantic feature for that dataset? The authors are highly encouraged to replace the visualization methods in Figure 4 with those in Figure 1 to clearly demonstrate the effects of each layer through multiple branches of each generated image, allowing readers to better understand which representations are controlled by previous fixed layers and which are affected by the remaining layers. Additionally, it would be very helpful if the authors could relate the observations from the qualitative results to the quantitative results in Table 1 and demonstrate whether they align with each other.

Second, the authors claim that the proposed framework outperforms existing methods on most datasets. However, it seems that many improvements are marginal. I appreciate the authors’ efforts in reporting the variance of performance in Table 2, but it is necessary to conduct a statistical test and report the p-value to demonstrate whether these marginal improvements are statistically significant.

**Essential References Not Discussed:**

N/A

**Experimental Designs Or Analyses:**

Yes, I checked all the qualitative and quantitative results. I believe the experiments are well-designed, but the analysis is not sufficient to support the key claims of this paper, as mentioned above in "Claims And Evidence".

**Methods And Evaluation Criteria:**

Yes, I believe most of the methods make sense and are suitable, while the qualitative results in Figure 4 can be improved by adapting the methods in Figure 1, as mentioned above.

**Other Comments Or Suggestions:**

# after rebuttal
I thank the authors for the response. Most of my concerns have been addressed, and I have increased my score accordingly.

**Other Strengths And Weaknesses:**

N/A

**Questions For Authors:**

Q(1): I look forward to the authors’ response to my concerns about the qualitative results, as described in the “Claims and Evidence” section, which will have the highest impact on my future decision.

Q(2): It would be very helpful if the authors could clarify my confusion listed in the “Theoretical Claims” section.

**Relation To Broader Scientific Literature:**

The proposed hierarchical model is novel and promising for INR works, which is a solid research direction worth investing in. However, the significant advantages and improvements of the current version of the proposed framework remain somewhat questionable compared to the baselines, as evidenced by the results.

**Theoretical Claims:**

TW (1): In line 188, how is $h^l$ computed? Is it obtained by sampling from a component of the mixture or by computing the average?

TW (2): Figure 2, as the main figure for the proposed framework, lacks sufficient captions to explain the details in the figure. For example, what does the transparency of noise indicate? What does the shade of color indicate? What is the relationship between each “latent” and the mixture of experts?

TW (3): In line 242, what does $||ε, ε_{θ}()||^2$ mean? Did the authors mean to write $||ε - ε_{θ}()||^2$, as used in diffusion models?

---

> ### Author Rebuttal · Authors · 2025-03-31
>
> We thank reviewer ZbmH for the valuable feedback.
>
> ### **W1. Qualitative results.**
>
> Thank you for the valuable feedback. We agree that earlier layers exhibit more visible influence in Figure 4. This is expected in our hierarchical design—deeper layers naturally introduce finer, subtler variations since prior layers are fixed. Regarding vehicle color in the NeRF dataset: it is not controlled by a specific layer but emerges from all layers jointly. This is because NeRF INRs jointly regress RGB and density at the output, causing color to be decoded at the final stage and entangled with spatial attributes, rather than controlled by a specific layer.
>
> We appreciate your suggestion to adopt Figure 1’s visualization style. However, Figure 4 was designed to showcase different conditional chains per row, and we cannot revise it due to rebuttal constraints. Nevertheless, please note that some layer effects are visible and aligned with Table 1—for example, in the NeRF samples, a spoiler appears in the 3rd and 4th samples of the first row, with layer 3 fixed in the latter, supporting its role in controlling that attribute.
>
> ### **W2. Marginal improvements.**
> Thank you for the suggestion. Our primary goal is not to optimize reconstruction/generation quality alone, but to introduce a framework that enables hierarchical control—a key capability missing in prior generative INR methods such as mNIF and Functa.
>
> As shown in Table 2, CHINR achieves notable improvements in evaluation metrics (e.g., PSNR/FID/SSIM) over most baselines except mNIF. However, CHINR exhibits significantly less memorization than mNIF, indicating better generalization. Furthermore, we aim to demonstrate that the conditional chain enables controllable generation without sacrificing quality—rather than to show it outperforms all models on every metric.
>
> ### **TW1. How is $h^l$ computed.**
> $h^l$ is an instance-specific latent at layer $l$, which is mapped to a gating vector that weighted averages the experts at layer $l$. To compute $h^l$, we use meta-learning or auto-decoding (Section 3.2) by initializing it around zero and optimizing it for each data instance.
>
> ### **TW2. Figure 2 meaning.**
> Thanks for pointing this out. We will make the captions clearer to explain this figure. Specifically, the transparency of noise refers to the noise schedule in the diffusion process, where the noise gradually dominates in the forward process. We shade the color of latents to distinguish latents from different layers. The latent at each layer is mapped to a gating vector that weighted averages the mixture of experts at that layer.
>
> ### **TW3. $||\epsilon,\epsilon_\theta()||^2$ in Line 242.**
> Thanks for pointing this out. It should be $||\epsilon - \epsilon_\theta()||^2$ as used in diffusion models. We will correct this in the final version.

---

### Official Review · Reviewer_aYYY · 2025-03-13

**Overall Recommendation:** 4

**Summary:**

The paper proposes a framework for controllable data generation using hierarchical implicit neural representations (INRs).
It models conditional dependencies across layers in the parameter space to improve control over the generation process.

**Claims And Evidence:**

The paper presents clear evidence to support the claims.

**Essential References Not Discussed:**

None

**Experimental Designs Or Analyses:**

The paper presents valid quantitative and qualitative experiments. Table 1 and 2 show the superior performance over existing methods, i.e. Functa, mNIF, GEM, and GASP. The analyses in Section 4.3 are intuitive. The ablation study demonstrates the effectiveness of conditional modeling.

**Methods And Evaluation Criteria:**

The proposed method and evaluation criteria are reasonable.

**Other Comments Or Suggestions:**

No

**Other Strengths And Weaknesses:**

Strength:
- The paper introduces a first-of-its-kind, hierarchical way to model INRs by incorporating
layer-wise conditional dependencies, allowing fine-grained control over semantic generation.
- The introduction of LOE greatly enhances the model’s ability to generate diverse semantics.
The expert-sharing mechanism also improves parameter efficiency.
- The framework is tested comprehensively across multiple modalities, including images, 3D
objects, and motion sequences.

Weakness:
- While the model enforces a hierarchical structure, the exact meaning of each layer’s latent representation is not explicitly defined. Users must manually inspect outputs to understand how different latents influence semantic factors. This lack of predefined interpretability might make fine-grained control difficult in real-world applications.
- Compared to state-of-the-art generative models such as StyleGAN or diffusion models,
CHINR appears to be restricted to relatively simple image contents and object representations. How does the model generalize to more complex data?
- The proposed framework introduces a multi-stage training pipeline involving meta-learning, auto-decoding, and hierarchical diffusion modeling, which is computationally intensive. A breakdown of training/inference time and resource requirements would be useful.

**Questions For Authors:**

No

**Relation To Broader Scientific Literature:**

The key contribution of the paper is the controllability of data generation with INRs. The proposed CHINR uses hierarchical latent vector and layer-of-experts to represent data instance while prior methods, e.g. Functa, and mNIF, use flat latent vector and MOE to represent data. The CHINR proposed hierarchical conditional diffusion model to model the conditional distribution, while prior methods model the joint distribution of flat latents.

**Theoretical Claims:**

The theoretical claims are correct and consistent. Equations (1), (2), and (3) discuss the foundation of hierarchical property of INR parameters. Equations (4) and (5) presents how to formulate the hierarchy modeling with diffusion models.

---

> ### Author Rebuttal · Authors · 2025-03-31
>
> We thank reviewer aYYY for the constructive feedback.
>
> ### **W1. Lack of predefined interpretability for each layer.**
>
> Thank you for the thoughtful comment. CHINR is designed to align the hierarchical structure of INRs with semantic abstraction, allowing each layer to control different levels of detail. While the semantics of each layer are not predefined (they are learned freely in Stage 1), the progressive control in Figure 4 and latent composition in Figure 6 demonstrate consistent layer-wise influence, suggesting that meaningful hierarchical semantics naturally emerge through training. Incorporating attribute supervision into Stage 1 to enforce predefined semantics at each layer is a promising direction to improve interpretability and fine-grained control in practical applications.
>
> ### **W2. Generalize to more complex data.**
>
> 1. Thank you for your insightful comments. Compared with other state-of-the-art methods such as StyleGAN, the data instance size represented by CHINR is significantly smaller (e.g., $5 \times 64$ latent vector). Additionally, the LoE utilizes a five-layer neural network, which inherently limits its generation quality. The primary advantage of using implicit neural representations (INRs) lies in their ability to represent data at arbitrary resolutions, rather than in achieving high reconstruction fidelity.
>
> 2. To generalize to more complex data, two main directions can be pursued: (1) enhancing the capacity of the INR, and (2) refining the meta-learning pipeline. For the first approach, dividing each data instance into smaller patches can significantly reduce the burden on the INR, allowing it to better capture local structures.
> For the second, existing meta-learning pipelines typically initialize the latent vector from a Gaussian distribution and optimize it over just three steps. This process can be improved by enabling more efficient optimization over a greater number of steps, thereby enhancing performance and adaptability.
>
> ### **W3. Training/inference cost**
>
> Thanks for your suggestion. We analyze the resource consumption of CelebA experiments in the following. All experiments were conducted on an RTX 3090 GPU with a batch size of 8 for both training and inference.
>
> *Training cost for Stage 1 and 2*:
> |         | Time | Memory |
> |---------|------|--------|
> | Stage 1 | 50h  | 4.3GB  |
> | Stage 2 | 28h  | 18.9GB |
>
> *Inference cost of CHINR, generating 1000 samples following HCDM*:
>  | Time | Memory |
> |------|--------|
> | 2h  | 8.4GB |

---

### Official Review · Reviewer_N8d9 · 2025-03-14

**Overall Recommendation:** 4

**Summary:**

The paper introduces CHINR, a framework for controllable data generation using hierarchical neural representations. It addresses limitations of existing generative INR approaches that fail to capture hierarchical structures in data, leading to limited control over generation. The method consists of two stages: Stage-1 constructs a Layers-of-Experts network where each layer has its own latent vector for disentangled representations, and Stage-2 introduces a Hierarchical Conditional Diffusion Model to capture dependencies across layers for controllable generation. The framework enables hierarchical control over generated content at various semantic granularities. Experiments across different modalities show improved generalization and controllability compared to existing methods.

## update after rebuttal
I hope the authors could refine the larger dataset problem in future revisions. As a concensus is reached among reviewers, I will keep my score.

**Claims And Evidence:**

The proposed CHINR achieves outstanding performance. This is verified by experiments.

**Essential References Not Discussed:**

No from my knowledge.

**Experimental Designs Or Analyses:**

I found no issues regarding experimental designs or analysis.

**Methods And Evaluation Criteria:**

The paper proposes LCDM and LoE as methods. The benchmarks are CelebA-HQ, ShapeNet, and SRN-Cars, which are commonly-used ones.

**Other Comments Or Suggestions:**

Layout problem: the font size of legend and ticks in Fig. 4 and 7 are too small. The authors should consider revising these figures.

**Other Strengths And Weaknesses:**

Strengths:
1. Novelty: hierarchical control is achieved, different from previous works.
2. Comprehensive evaluation: the multi-modal experiments have been conducted to validate the method in various scenarios.

Weaknesses:
Scalability to larger datasets, as is mentioned in "Conclusions".

**Questions For Authors:**

No further questions.

**Relation To Broader Scientific Literature:**

While existing INR methods fail to leverage the hierarchy of semantic abstraction, this paper introduces hierachical control in generative INRs.

**Theoretical Claims:**

No theoretical claims are involved in this paper.

---

> ### Author Rebuttal · Authors · 2025-03-31
>
> We thank reviewer N8d9 for the valuable comments.
>
> ### **W1. Scalability to larger datasets.**
>
> Thank you for pointing this out. As discussed in the conclusion, scaling CHINR to larger datasets is a known challenge. While CHINR demonstrates the core idea of hierarchical control through INR parameter modeling, extending it to more complex data can be achieved by enhancing the INR representation capacity and employing more efficient training approaches. For example, localized solutions such as patch-wise or spatial-adaptive modulation methods [1,2,3] can help INRs better capture local structures and improve scalability.
>
> Thanks for pointing out the layout problem of Fig. 4 and 7. We will increase the font size of those figures.
>
> ### **Reference**
> [1] Wang, Peihao, et al. Neural implicit dictionary learning via mixture-of-expert training. ICML, 2022.
> [2] Bauer, Matthias, et al. Spatial functa: Scaling functa to imagenet classification and generation. 2023.
> [3] Park, Dogyun, et al. DDMI: Domain-Agnostic Latent Diffusion Models for Synthesizing High-Quality Implicit Neural Representations. ICLR, 2024.

---

### Official Review · Reviewer_T7ed · 2025-03-15

**Overall Recommendation:** 4

**Summary:**

The paper presents a novel method, CHINR, for controllable generative INR by exploiting the hierarchy structure in parameters. The authors employ a Layers-of-Experts (LoE) network to encode data with layer-wise latents and propose a Hierarchical Conditional Diffusion Model (HCDM) to learn conditional dependency in layers. Experiments show that CHINR enables precise control at different granularities during generation.

**Claims And Evidence:**

The claims are supported by clear evidence.

**Essential References Not Discussed:**

None

**Experimental Designs Or Analyses:**

The experimental designs and analyses are convincing. Figure 4 clearly presents controllable data generation, and Table 1 presents accordingly quantitative evidence. Figure 6 intuitively explains why the CHINR works.

**Methods And Evaluation Criteria:**

The proposed CHINR makes sense in solving controllable data generation. The evaluation criteria such as success rate and FID are convincing.

**Other Comments Or Suggestions:**

None

**Other Strengths And Weaknesses:**

Strength:
- The observation of connection between hierarchy of semantics and model parameters is interesting and inspiring, and the idea is well-motivated.
- The proposed diffusion model is a novel method to model the hierarchical dependency within parameters.
- The paper is well-written and structure is straightforward. The mathematical formulations appear correct.
- The empirical results show strong evidence of successful controllability across various modalities. The paper presents thorough analysis that semantics are disentangled in parameter space.

Weakness:
- Since the parameters are generated with a conditional chain, it has chances getting out-of-distribution samples. The out-of-distribution causes erroneous parameters, which will accumulate if occurred in early layers. How do you solve this problem?
- The binary condition length must be carefully tuned on each dataset to balance controllability and generalization. Can the authors explain more about the choice of binary condition?
- How does the model generalize to data without inherent frequency hierarchy, e.g. text?
- The stage-2 is trained with reconstructed data of stage-1, how does the reconstruction quality affect the conditional modeling in stage-2? Can stage 2 learn compatible conditional chain if the reconstruction quality is low?

**Questions For Authors:**

None

**Relation To Broader Scientific Literature:**

The paper explores an orthogonal direction to existing INR literature by focusing on the controllability of data generation. It argues that methods such as Functa, mNIF, and GASP process the latent modulation vector indiscriminately, and instead proposes modeling this vector as a conditional chain to enable hierarchical control.

**Theoretical Claims:**

The theoretical claims are correct and clearly explained. We observed that Equation (2) in main paper is grounded in explaining the representation ability of INR, while Equation (3) models corresponding hierarchical representations.

---

> ### Author Rebuttal · Authors · 2025-03-31
>
> We thank reviewer T7ed for the thoughtful comments.
>
> ### **W1. Out-of-distribution samples in conditional chain.**
>
> Thank you for your insightful feedback. We address error accumulation during both the training and inference phases.
>
> In the training phase, we begin by teaching the model to capture inter-layer dependencies, using ground truth layer-wise latents (obtained from Stage 1) as conditions. Subsequently, we fine-tune the model to construct a coherent conditioning chain by progressively incorporating generated latents as conditions.
>
> During inference, we further mitigate potential degradation by adjusting the standard deviation of the input noise in the diffusion process. This reduces the likelihood of generating outlier latents that could compromise output quality.
>
> ### **W2. Binary condition.**
>
> We choose to use binary condition for the following reasons:
>
> 1. With fewer bits to represent data, the model is less likely to memorize every detail of the training data. Instead, it must extract the core features that are most useful/general across many examples. This acts as a form of regularization.
>
> 2. As the model processes the input data, it learns to assign similar items to the same quantized value. Over time, these discrete codes become representative of certain semantic concepts or clusters, which correspond to the inherent structure of the data.
>
> ### **W3. Generalize to data without inherent frequency hierarchy.**
>
> Thank you for your interesting question. CHINR leverages the frequency-based representational hierarchy inherent in INR structure, which aligns well with data like images and 3D shapes with spatial frequency hierarchy. The current formulation may not be directly applicable for modalities like text that don't follow such a frequency hierarchy. However, CHINR’s core idea—modeling hierarchical latent dependencies across network layers—may still be adapted by identifying alternative structures (e.g., syntactic or semantic hierarchies) relevant to the modality.
>
> ### **W.4 Reconstruction quality affects Stage 2**
>
> The sensitivity of hierarchical control to quality variations is evident in two key aspects:
>
> 1. Reconstruction Quality in Stage 1: The effectiveness of hierarchical control in Stage 2 is bounded by the reconstruction quality achieved in Stage 1. If the proposed LoE fails to learn hierarchically structured latents during Stage 1, it limits the controllability during the generation in Stage 2. Our findings show that when the PSNR on CelebA-HQ exceeds 25, no visually noticeable distortions are observed, and hierarchical control performs reliably.
>
> 2. Quality of Ground-Truth Data: The controllability is also constrained by the quality of the ground-truth data used in Stage 1. Higher noise levels hinder the learning of conditional dependencies. In the extreme case where the data is entirely noise, the resulting latents lack meaningful structure. As the noise in the ground-truth data increases, both reconstruction and generation quality progressively degrade.

---

### Official Review · Reviewer_XvvS · 2025-03-16

**Overall Recommendation:** 4

**Summary:**

The paper introduces a novel framework to capture hierarchical data semantics with implicit neural representations, enabling improved control over data generations. The framework is structured in two stages. For the first stage, a layer-of-expert (LOE) architecture is employed to capture general semantics with shared experts and distinct semantics with latent vectors. For the second stage, a hierarchical conditional diffusion model (HCDM) is employed to learn the distribution of latent vectors. The HCDM models inherent hierarchical structure as conditional dependencies between layer l and layers <l. The framework is evaluated on four different domain data, i.e. images, point clouds, neural radiance fields, and motions, and presents superior performance in controllability and reconstruction ability.

## update after rebuttal
I will keep my ratings since most of my concerns are solved.

**Claims And Evidence:**

Each claimed contribution is supported by clear evidence.

**Essential References Not Discussed:**

None

**Experimental Designs Or Analyses:**

The experimental designs are valid in proving the controllability achieved through the hierarchical approach. The analyses in section 4.3 are convincing and show evidence of disentangled semantic. The only issue would be the scale of data as CHINR is currently evaluated in small-scale datasets.

**Methods And Evaluation Criteria:**

The proposed hierarchical approach makes sense. It aligns INR structure with data semantic for layer-wise control. The evaluation criteria (e.g., Table 1) are reasonable in showing the controllability of CHINR.

**Other Comments Or Suggestions:**

See the weaknesses part.

**Other Strengths And Weaknesses:**

Strengths:

1. The idea of bridging INR weight hierarchy and semantic hierarchy is novel, and the layer-wise generation effectively models this hierarchy.

2. The experiments show CHINR’s controllability in different datasets, validating its hierarchical approach across multiple domains.

3. The paper is well written. The motivation, methodology, and experiments are clearly presented.

Weaknesses:

1. Despite performing well on small datasets like CelebA-HQ, CHINR’s scalability to larger and more complex data (e.g., higher-resolution images or shapes) is unclear. Adding more experts or layers may hinder training convergence and raise inference costs. In addition, longer conditional chains are more prone to error accumulation, affecting generation quality.

2. The reliance on meta-learning may limit generalization to more diverse data patterns. Since latents are initialized from a small distribution for fast adaptation, they struggle to capture broader variations.

3. CHINR allows attribute variation at different levels but lacks a mechanism for targeted modification (e.g., changing pink lips to red). This limits its use in real-world applications such as image and shape editing, where targeted adjustments are essential.

**Questions For Authors:**

See the weaknesses part.

**Relation To Broader Scientific Literature:**

CHINR aims at bridging the weight space structure and data semantics for controlled generation, which is new to the INR literature. The idea of hierarchical/progressive data generation has been widely explored in a broader literature (e.g., GANs, VAEs). The difference is that CHINR models this hierarchy in network parameter space instead of data feature space.

**Theoretical Claims:**

The theoretical claims seem correct. Equation 2 explains the hierarchical structure of INR, which serves as the foundation of the hierarchical approach.

---

> ### Author Rebuttal · Authors · 2025-03-31
>
> We thank reviewer XvvS for the valuable feedback.
>
> ### **W1. Scalability to larger datasets**
>
> Thanks for raising this scalability concern. The CHINR framework focuses on establishing the connection between INR parameters and data semantics for controllable generation. While our experiments use smaller datasets, the core framework is generalizable to larger-scale data as long as this fundamental connection holds. To handle increased complexity, more efficient learning methods should be employed to handle more complex data patterns.
>
> Beyond simply adding more layers or experts, we can adopt localized solutions such as patch-wise or spatial-adaptive modulation, as explored successfully in [1, 2]. This reduces the burden of a single set of INR parameters to represent the whole data and allows the model to capture local structures, improving scalability and convergence without significantly increasing inference cost.
>
> ### **W2. Reliance on meta-learning**
>
> Thanks for pointing out this thoughtful concern. We agree that meta-learning requires a shared initialization for fast adaption, which works well for data with consistent structure (e.g., faces) but may struggle with more complex data. This is an intrinsic issue of the meta-learning method. To address this, one can allow more inner-loop updates at the cost of increased training time. It is also possible to incorporate more adaptive initialization or weight update strategies [3, 4] to better handle data variability. Alternatively, we can use auto-decoding as done in the NeRF experiments, which avoids the need for fast adaptation and may offer better generalization in such cases.
>
> ### **W3. Lack of targeted modification mechanism**
>
> Thank you for the insightful comment. CHINR is designed to leverage the hierarchical structure of INR parameters for layer-wise semantic control, rather than direct attribute editing. Although it doesn't currently support direct modifications like changing lip color, it can be extended with attribute supervision (in Stage 1) or latent manipulation to enable targeted edits—an exciting direction for future work.
>
> ### **Reference**
> [1] Bauer, Matthias, et al. Spatial functa: Scaling functa to imagenet classification and generation. 2023.
>
> [2] Park, Dogyun, et al. DDMI: Domain-Agnostic Latent Diffusion Models for Synthesizing High-Quality Implicit Neural Representations. ICLR, 2024.
>
> [3] Wang, Ruohan, et al. Structured prediction for conditional meta-learning. Neurips, 2020.
>
> [4] Baik, Sungyong, et al. Meta-learning with adaptive hyperparameters. Neurips, 2020.

---

> > ### Comment · Reviewer_XvvS · 2025-04-07
> >
> > Solved most of my concerns. I will keep my positive ratings.

---

### Decision · Program_Chairs · 2025-05-01

**Decision:**

Accept (poster)

**Comment:**

The paper introduces a genuinely novel and well-motivated approach to controllable generation within the INR framework. The core contribution – linking parameter hierarchy to semantic control via conditional modeling – is significant and opens up new possibilities for generative INRs. The method is technically sound and supported by extensive experiments demonstrating the key claim of hierarchical controllability across diverse data types. While limitations regarding scalability, interpretability, and targeted editing exist, they are reasonably acknowledged as scope limitations or avenues for future work. The strong consensus among reviewers, bolstered by effective author rebuttals, supports acceptance. This work is a valuable addition to the literature on generative modeling and implicit neural representations.